# MIXTURE OF THOUGHTS: LEARNING TO AGGREGATE WHAT EXPERTS THINK, NOT JUST WHAT THEY SAY

## ABSTRACT

Open-source Large Language Models (LLMs) increasingly specialize by domain (e.g., math, code, general reasoning), motivating systems that leverage complementary strengths across models. Prior multi-LLM approaches either (i) *route* a query to one or a few experts and generate independently, (ii) *aggregate* outputs from each model via costly multi-turn exchanges, or (iii) *fuse* weights into a single model—typically requiring architectural homogeneity. We introduce *Mixture of Thoughts* (MoT), a simple method for *latent-level collaboration* among heterogeneous experts under a global routing scheme. For each query, a lightweight router selects top-$K$ experts and designates a primary expert; uniformly placed *interaction layers* project hidden states into a shared latent space where the primary expert performs cross-attention over its active (selected) peers. Pre-trained experts remain frozen; only the router and the lightweight interaction layers are trained with a novel joint training objective that improves both the expert selection and inter-expert collaboration. Across five in-distribution (ID) and three out-of-distribution (OOD) benchmarks, MoT surpasses the current routing and aggregation-based state-of-the-art, AVENGERS, by +0.38% and +2.92%, respectively. Further, MoT significantly outperforms the best-performing single model. It achieves this with single-pass inference, runtime comparable to routing baselines, and none of the overheads of iterative aggregation. MoT offers a simple latent-space mechanism for combining heterogeneous LLMs, a practical step toward broader multi-LLM collaboration. Our code is publicly available at `https://anonymous.4open.science/r/mot-5B4B/%7D`.

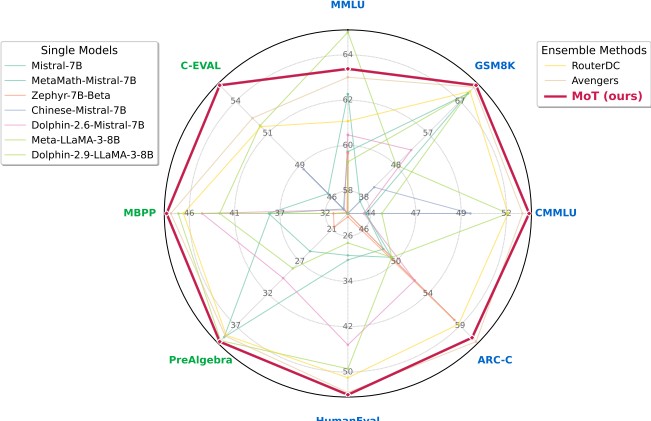

Figure 1: Accuracy radar plot across OOD and ID tasks, showing MoT outperforming base models and state-of-the-art routing baselines. Scores are normalized to a $0 - 100$ range for visualization.

## 1 INTRODUCTION

Large Language Models (LLMs) have reshaped applications in mathematics, coding, multimodal reasoning, and language understanding [61]. Open-source LLMs (e.g., Mistral [25], LLaMA 3 [12],

LLaMA 4 [37], Qwen 3 [53]) are competitive, yet closed-source systems (GPT-5 [40], Claude Opus 4.1 [1], Grok 4 [50], Gemini 2.5 [29]) still lead on widely used and highly competitive benchmarks, including MMLU [18], GSM8K [10], BBH [44], HumanEval [5], ARC-Challenge [9], GPQA [42], MMMU/MMMU-Pro [56, 57], SWE-Bench Verified [27, 8], and Humanity's Last Exam [41].

Specialized open-source models excel in particular skills—math [54], code [34], or common-sense [45]—suggesting complementary expertise that could be combined. Prior multi-LLM efforts do so via: (i) *selective routing* to one or a few experts [6, 35, 63, 39, 15, 20, 28, 26, 60, 58, 23], sometimes with ensemble sampling [58]; (ii) *response-level collaboration* (mixture-of-agents) [48, 31, 33, 4]; and (iii) *parameter-space fusion* (merging, continual learning, distillation) [51, 55, 14, 21, 59, 52, 47]. These strategies are coarse-grained and/or compute-heavy, often preserving only narrow skills and weakening generalization [7].

**Limitations.** Learned routing (e.g., ROUTERDC [6]) maps each query to a single expert, binding performance tightly to the chosen expert without any cross-model interaction. Multi-expert routing (e.g., AVENGERS [58], current state of the art) relaxes this by selecting multiple experts, but combines their outputs through ensembling via sampling and voting [49, 3], rather than true collaboration. While response-level collaboration (e.g., MOA [48]) aggregates final outputs (responses) from multiple models, it does so at the cost of multi-turn iterative exchanges between models. Parameter fusion approaches (e.g., TIES-MERGING [51], FUSELLM [47]) require architectural homogeneity and collapse specialization into a single set of weights, removing per-query adaptivity.

**Mixture of Thoughts (MoT).** All prior paradigms lack fine-grained interaction in the latent space, restricting collaboration to outputs, weight fusion, or naive ensembling. To address this gap, we propose *Mixture of Thoughts* (MOT), a method for *latent-level collaboration* among heterogeneous experts under a global router. For each query, a lightweight trainable router selects the top-$K$ experts and designates a *primary* expert. Within each expert, we uniformly place *interaction layers* composed of lightweight adapters (projections) and a cross-attention module. At these layers, all active experts' hidden states are projected into a shared latent space and integrated into the primary expert via cross-attention—enabling fine-grained collaboration while keeping each expert's backbone frozen. A joint training objective optimizes both the router and interaction layers. MOT performs a *single forward pass* through selected experts, achieving routing-like efficiency with richer representational capacity than single- and multi-model routing, response-level aggregation, and parameter fusion, while supporting fully heterogeneous experts with dynamic, per-query adaptive collaboration (Fig. 2).

Unlike prior work that aggregates only at the output (*'say'*), MOT enables experts to integrate hidden representations (*'thoughts'*) via interaction layers.

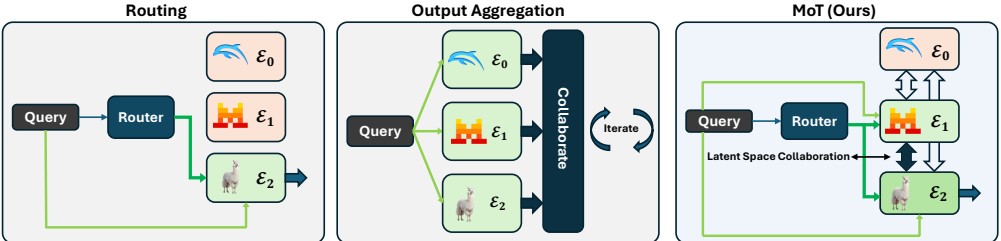

Figure 2: Prior paradigms (routing, output aggregation) vs. our MOT schematic. Parameter fusion schematic not shown as it does not support heterogeneous models. Dark green lines are for selected experts.

We evaluate MOT on language understanding, code generation, and mathematical reasoning under in-distribution (ID) and out-of-distribution (OOD) settings. MOT attains state-of-the-art performance among open-source multi-LLM methods: it surpasses the strongest single model (Dolphin-2.9-LLaMA-3-8B) by **+10.95**% (ID) and **+9.03**% (OOD), improves over ROUTERDC by **+3.40**% (ID) and **+4.52**% (OOD), and exceeds AVENGERS by **+0.38**% (ID) and **+2.92**% (OOD), with comparable wall-clock cost (51.8 vs. 51.3 minutes). MOT is robust to the number of selected experts, expert drop-out, the number of interaction layers (and their placement), and shared latent-space dimensionality. Compared to Avengers [58], a strong routing-based baseline, MoT consistently matches or improves performance on four of five in-domain tasks and all three out-of-domain tasks, while adding only

about 1–3% trainable parameters on top of a 45.75B frozen expert pool and maintaining essentially identical inference latency. This positions MoT as a method that requires only a small amount of offline training while delivering improved accuracy under the same test-time budget.

| Capability | R | C | F | MoT |
|---|---|---|---|---|
| Latent-level collaboration | × | × | × | ✓ |
| Cost-efficient (single-pass) | ✓ | × | ✓ | ✓ |
| Supports heterogeneous experts | ✓ | ✓ | × | ✓ |
| Per-query collaborative adaptation | × | ✓ | × | ✓ |
| Beyond output-level fusion | × | × | ✓ | ✓ |

**Representative works: R** (routing): [6, 35, 63, 39, 28, 26, 15, 20, 60, 58, 23]; **C** (response-level collaboration): [48, 31, 33, 4, 58]; **F** (fusion): [51, 55, 59, 52, 14, 21, 47];

**Contributions.** (i) MoT: combines a lightweight global router with *latent-level* interaction layers, enabling per-query collaboration among heterogeneous, frozen experts. (ii) A joint training objective optimizes the router and interaction layer parameters. (iii) *Single-pass* multi-expert inference yields lower overhead than response-level collaborative methods (e.g., MoA) and greater representational capacity than routing-only baselines (e.g., ROUTERDC, AVENGERS). (iv) Consistent ID/OOD gains: **+3.40**% / **+4.52**% over ROUTERDC, and **+0.38**% / **+2.92**% over AVENGERS, at comparable wall-clock cost. (v) Robust to expert count, drop-out, interaction depth/placement, and shared latent size; preserves specialization while enabling dynamic query-level collaboration.

# 2 RELATED WORK

**Routing (Selection-Based).** Routing-based integration methods use a learned or heuristic router to select one (or a small subset) of experts per query. ROUTERDC [6] trains a parameter-efficient dual-contrastive router; ZOOTER [35] employs zero-shot prompt scoring; EMBEDLLM [63] embeds queries and models in a joint space; ROUTELLM [39] learns task-specific gates; and GRAPHROUTER [15] models skill relationships over a graph. ROUTERBENCH [20] evaluates routers across tasks, and [28] proposes task-agnostic, transferable routers. Multi-model variants (LLMBLENDER [26], MODEL-SAT [60], AVENGERS [58], DEEPEN [23]) select top-$K$ experts and combine outputs via ensembling (sampling & voting, [3, 49]). These approaches typically fuse only *final* generations, discarding intermediate states and making performance tightly depend on the routed experts' outputs. In contrast, MoT preserves the efficiency of selection while enabling *latent*-level exchange among the top-$K$ routed experts, allowing the primary expert to efficiently aggregate token-level information across experts, in a single-pass.

**Response-Level Collaboration.** Agentic systems coordinate multiple LLMs to iteratively refine responses: MOA [48] orchestrates role-specialized agents, SPARSE-MOA [31] activates a subset per turn, SELF-MOA [33] adds self-reflection, and SYMBOLIC-MOE [4] blends symbolic reasoning. While effective, these methods incur high inference cost from multi-round generation, interacting only at the response level. MoT attains collaborative benefits without iterative turns by sharing intermediate representations: active experts contribute projected hidden states that the primary expert cross-attends to within interaction layers, improving generation quality at single-pass inference cost.

**Parameter Fusion / Distillation.** Weight-space integration merges experts into a single model, often assuming architectural homogeneity: TIES-MERGING [51] and DARE [55] merge parameters; MODEL-SWARMS [14], LORAHUB [21], and GENOME [59] compose specialists via incremental or evolutionary search; [52] discusses homogeneity and scalability constraints of these approaches. Distillation methods such as FUSELLM [47] transfer multi-expert knowledge via generation super-vision. These yield a single efficient model but collapse collaboration into static weights, diluting specialization and removing per-query adaptivity. MoT instead keeps all backbones frozen and heterogeneous, adds only lightweight projectors and cross-attention, and adapts collaboration per query through global routing and token-level latent integration.

**Our work (MoT).** Prior work has centered on either routing queries to the most suitable expert, aggregating outputs through costly multi-turn interactions, or collapsing models via fusion and distillation that require architectural homogeneity and sacrifice diversity. In contrast, MoT introduces

latent-space collaboration across heterogeneous experts, retaining individual model specialization while enabling efficient single-pass inference.

## 3 METHOD

We now describe the proposed *Mixture of Thoughts* (MoT) framework. Section 3.1 provides a high-level overview of the architecture. Section 3.2 details the global routing mechanism that selects a sparse subset of expert models from the given pool of LLMs. Section 3.3 introduces the interaction layers that enable latent-space collaboration among routed experts. Section 3.4 summarizes the training strategy, and Section 3.5 describes the inference procedure.

### 3.1 OVERVIEW

MoT integrates multiple pre-trained decoder-only language models to enable collaborative reasoning without modifying their backbone parameters. The system maintains $M$ frozen experts $\{\mathcal{E}_0, \ldots, \mathcal{E}_{M-1}\}$, which may differ in size, architecture, training corpus, or domain specialization. For each input, a lightweight global router activates a sparse Top-$K$ subset of experts, designating the highest-scoring one as the *primary expert* responsible for output generation. Collaboration occurs through lightweight interaction layers that operate in a shared latent space, allowing the primary expert to integrate hidden representations from its peers via cross-attention. Unlike multi-turn approaches (e.g., [48, 58]), MoT requires only a single generation pass, offering both efficiency and fine-grained inter-expert collaboration.

To align experts of different depths (number of layers), we partition each $\mathcal{E}_m$ with $L_m$ transformer layers into $Q$ contiguous stacks (Fig. 3): $\mathcal{E}_m = \{S_m^0, S_m^1, \ldots, S_m^{Q-1}\}$ where $|S_m^q| = L_m/Q$. This enables $Q$ levels of interactions. At the end of each stack, routed experts project their hidden states into a shared latent dimension; the primary expert attends over these projected states and updates its representation before continuing with its forward pass. The backbones remain frozen, while only the router and the projection/attention layers are trained, adding minimal parameter overhead (see Appendix H).

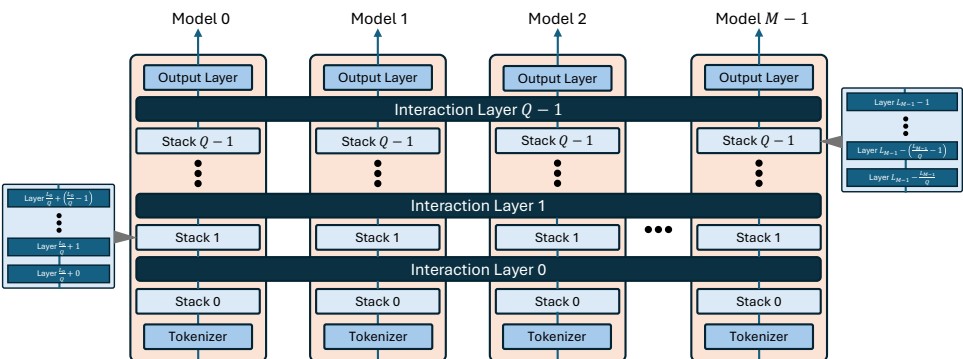

Figure 3: **Heterogeneous expert integration in MoT.** Each expert with $L_m$ layers is divided into $Q$ stacks, enabling $Q$ levels of interaction. After each stack, routed experts project hidden states into a shared latent space; the primary expert aggregates these via cross-attention before proceeding.

### 3.2 GLOBAL ROUTING

The global router (Fig. 4) selects a sparse subset of experts for each input, ensuring that only the most relevant models participate in collaboration. Given an input prompt $\mathbf{x}$, we first compute a fixed-dimensional embedding $\mathbf{z} = \text{PromptEncoder}(\mathbf{x}) \in \mathbb{R}^d$. A lightweight MLP router $r_\theta$ maps this embedding to expert relevance scores $\mathbf{s} = r_\theta(\mathbf{z}) \in \mathbb{R}^M$.

The active set of experts is obtained by taking the indices of the top-$K$ scores, $\mathcal{I}_{\text{active}} = \text{TopK}(\mathbf{s}, K)$, and the highest-scoring expert is designated as the *primary expert*, $m^* = \arg\max_{m \in \mathcal{I}_{\text{active}}} s_m$. The active experts proceed with their forward pass in parallel, with latent space collaboration via interaction layers at the end of each stack (Fig. 3). The final output tokens are generated via the primary expert alone – integrating information from its active peers, for a given query, via the interaction layers. We utilize a frozen sentence encoder (DeBERTaV3 [17]) to obtain $\mathbf{z}$, while the router $r_\theta$ is trained jointly with the interaction layers (see Sec. 3.4).

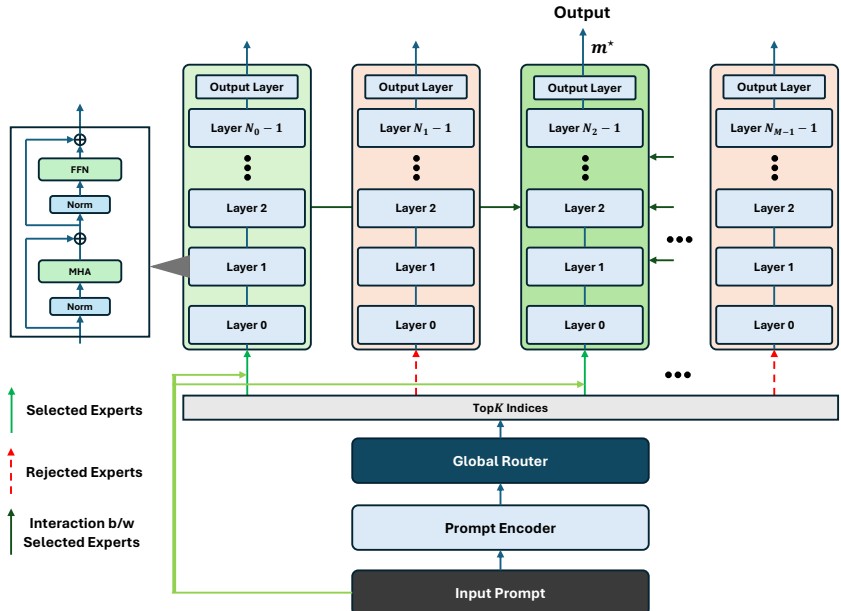

Figure 4: **Global router.** Each input prompt is encoded, scored against all experts, and the top-$K$ are activated. The highest-scoring expert is designated as the primary decoder responsible for output generation.

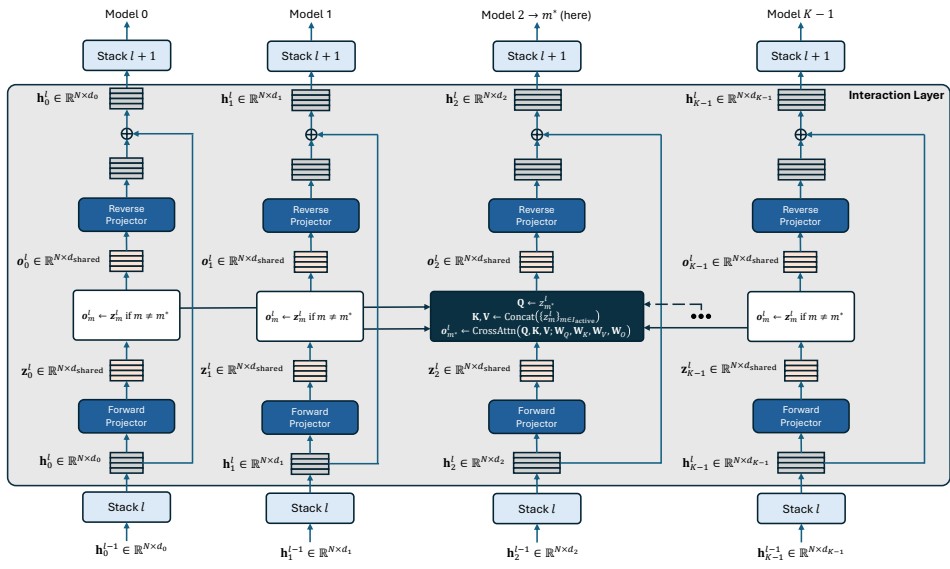

Figure 5: **Interaction layers.** Active experts project hidden states to a shared space; the primary expert integrates them via cross-attention, while non-primary experts pass through.

## 3.3 INTERACTION LAYERS

At the end of each stack $\ell \in \{0, \ldots, Q-1\}$, routed experts collaborate in a shared latent space (Fig. 5). Let $\mathbf{h}_m^\ell \in \mathbb{R}^{N \times d_m}$ be the hidden states of expert $m$ after stack $\ell$. Each active expert applies a ForwardProjector, a linear layer mapping into the shared dimension $d_s$: $\mathbf{z}_m^\ell = \mathrm{ForwardProjector}_m^\ell(\mathbf{h}_m^\ell)$. For the primary expert $m^*$, cross-attention integrates the projected states from all active experts. We set query$^\ell = \mathbf{z}_{m^*}^\ell$ and key$^\ell =$ value$^\ell = \mathrm{concat}\{\mathbf{z}_m^\ell \mid m \in \mathcal{I}_{\mathrm{active}}\}$. After projection via weights $\mathbf{W}_Q^\ell$, $\mathbf{W}_K^\ell$, and $\mathbf{W}_V^\ell$, we get the query ($\mathbf{q}^\ell$), key ($\mathbf{k}^\ell$), and value ($\mathbf{v}^\ell$) matrices, respectively. We compute cross-attention via:

$$\mathbf{A}^\ell = \mathrm{softmax}\left(\frac{\mathbf{q}^\ell(\mathbf{k}^\ell)^\top}{\sqrt{d_s}} + \mathbf{M}^\ell\right)\mathbf{v}^\ell, \qquad \mathbf{o}_{m^*}^\ell = \mathbf{A}^\ell \mathbf{W}_O^\ell.$$

In practice, we use multi-headed cross-attention [46] with $H = 8$ heads. $\mathbf{M}^\ell \in \mathbb{R}^{N \times (KN)}$ is a block-causal mask ensuring autoregressive visibility across tokens of all active experts (see Appendix C). The primary expert hidden states are updated via a ReverseProjector: $\mathbf{h}^\ell_{m^*} \leftarrow \mathbf{h}^\ell_{m^*} + \text{ReverseProjector}^\ell_{m^*}(\mathbf{o}^\ell_{m^*})$. For each non-primary active expert $m \neq m^*$, no cross-attention is computed. Instead, states pass through the projector pair to keep parameters trainable while avoiding extra compute: $\mathbf{h}^\ell_m \leftarrow \mathbf{h}^\ell_m + \text{ReverseProjector}^\ell_m(\mathbf{z}^\ell_m)$. Thus only the primary expert executes cross-attention, while all experts train their lightweight projectors. This enables frozen backbones to collaborate through a shared latent space at minimal cost. Final token generation is taken from the primary expert, while non-primary experts continue decoding to maintain diverse latent trajectories that feed subsequent interactions in each decoding step. The trainable parameters are the router $\Theta_{\text{router}}$, forward/reverse projectors $\Theta_{\text{proj}} = \{(\mathbf{F}^\ell_m, \mathbf{R}^\ell_m)\}_{m,\ell}$, and per-layer shared attention weights $\Theta_{\text{attn}} = \{(\mathbf{W}^\ell_Q, \mathbf{W}^\ell_K, \mathbf{W}^\ell_V, \mathbf{W}^\ell_O)\}_\ell$.

## 3.4 TRAINING

We train only the router and interaction layers while keeping all expert backbones frozen (Sec. 3.3). The main training objective is the standard autoregressive language modeling loss. To stabilize routing, we add two standard regularizers: (i) an entropy term on the pre-selection scores $\mathbf{s} = r_\theta(\mathbf{z})$ to encourage exploration [24, 36], and (ii) a load-balancing term that penalizes variance in expert activation frequencies across a batch, as in sparse MoE training [43, 13]. The full objective combines these components with nonnegative weights. Routing is discrete Top-$K$ in the forward pass. Gradients are obtained using a straight-through estimator: i.i.d. Gumbel noise is added to $\mathbf{s}$, a soft Top-$K$ distribution is computed, and backpropagation proceeds through the soft scores while the hard indices are used for expert routing [30]. Prior work has noted that instability in expert selection can degrade convergence and efficiency [11, 38]. To address this, we introduce a routing-consistency term that penalizes divergence between outputs obtained from two independently perturbed Top-$K$ selections for the same input. This encourages stability under small routing variations while keeping training aligned with discrete inference. Refer to Appendix B for specific details.

## 3.5 INFERENCE

Inference in MoT proceeds in two phases. In the *prefill* phase, the input prompt is routed to the top-$K$ experts selected by the global router. Each active expert processes its stacks in parallel, with interaction layers enabling the primary expert to integrate projected hidden states from all active experts via masked cross-attention, while non-primary experts only apply forward/reverse projections without attention. In the *decode* phase, the primary expert generates the next token at each step, while non-primary active experts also produce tokens independently to advance their hidden states. Collaboration continues in the interaction layers, where the primary expert performs cross-attention. Generation ends when the primary expert reaches end of sequence token or the maximum length is reached; if a non-primary expert halts earlier, its outputs are treated as padding and masked in subsequent steps. Interaction layers use lightweight projections and masked cross-attention with standard causal masking, and key–value (KV) pairs can be cached across all experts as in standard LLM inference. This makes MoT compatible with efficient KV caching while supporting collaborative generation. Further details are provided in Appendix C.

# 4 EXPERIMENTS

## 4.1 SETUP

**Models.** MoT routes among seven 7B to 8B decoder-only LLMs drawn from the Mistral and Llama-3 families, covering general, instruction-tuned, aligned, math-specialized, and Chinese-continued-pretraining variants (details in Appendix A, J). Our experimental setup closely follows prior work [6, 58]. All reported numbers are from a single run per configuration with a fixed random seed, following prior work in multi-LLM routing and collaboration [58, 6].

**Datasets.** Training uses the union of train splits from five in-distribution benchmarks—**MMLU** [18], **GSM8K** [10], **CMMLU** [32], **ARC-C** [9], and **HumanEval** [5]—with evaluation on their held-out tests. For GSM8K we use its default split; for the others we follow standard practice (from [6, 58]) with a random 70/30 train/test split. Out-of-distribution (OOD) generalization is assessed only on **PreAlgebra** [19], **MBPP** [2], and **C-Eval** [22].

**Baselines.** We compare to (i) training-free routers: *Voting* and *CosineClassifier*; (ii) clustering-based routers: *Avengers* [58] [1] and *LoraRetriever* [62]; and (iii) supervised routers: *RouterDC* [6] and *ZOOTER* [35]. All methods use the same expert pool and datasets for evaluation and training. Refer Appendix A for implementation details and Appendix J for training details.

## 4.2 RESULTS

**In-distribution (ID).** Table 1 reports accuracy on five ID benchmarks. MoT attains the best average score (60.53), outperforming the prior SOTA AVENGERS (60.30) by $+\mathbf{0.38}\%$ and the strongest single-model baseline (Dolphin-2.9-LLaMA-3-8B; 54.56) by $+\mathbf{10.95}\%$. Relative to ROUTERDC (58.54), MoT improves ID average by $+\mathbf{3.40}\%$. Per-task, MoT achieves the top result on MMLU, GSM8K, CMMLU, and HUMANEVAL (4/5 tasks, wrt routing & aggregation baselines), with a narrow second place on ARC-C. Despite introducing interaction layers, runtime remains comparable to AVENGERS (51.8 vs. 51.3 minutes on the same hardware).

Table 1: In-distribution results (accuracy %, higher is better). "Time (m)" is average end-to-end evaluation minutes under the shared setup.

| Method | MMLU | GSM8K | CMMLU | ARC-C | HumanEval | Avg | Time (m) |
|---|---|---|---|---|---|---|---|
| **Base models** | | | | | | | |
| Mistral-7B | 62.14 | 36.71 | 43.83 | 49.43 | 28.98 | 44.22 | 38.8 |
| MetaMath-Mistral-7B | 59.86 | 69.63 | 43.83 | 48.30 | 29.80 | 50.28 | 40.5 |
| Zephyr-7B-Beta | 59.81 | 33.00 | 42.82 | 57.95 | 22.04 | 43.13 | 40.6 |
| Chinese-Mistral-7B | 57.42 | 41.03 | 49.67 | 43.47 | 21.43 | 42.60 | 40.2 |
| Dolphin-2.6-Mistral-7B | 60.53 | 52.38 | 43.71 | 52.56 | 45.10 | 50.86 | 42.1 |
| Meta-LLaMA-3-8B | 64.59 | 47.76 | 51.77 | 49.43 | 26.73 | 48.06 | 41.2 |
| Dolphin-2.9-LLaMA-3-8B | 59.46 | 69.81 | 44.72 | 49.43 | 49.39 | 54.56 | 38.6 |
| **Ensembles / routers** | | | | | | | |
| Voting | 63.30 | 67.39 | 47.48 | 50.85 | 42.85 | 54.37 | 343.8 |
| CosineClassifier | 59.72 | 69.03 | 45.47 | 50.57 | 46.33 | 54.22 | 49.7 |
| ZOOTER | 60.48 | 66.69 | 45.27 | 53.13 | 44.29 | 53.97 | 47.3 |
| LoraRetriever | 63.33 | 66.63 | 51.77 | 57.10 | 40.00 | 55.77 | 46.2 |
| RouterDC | 61.07 | 70.32 | 51.77 | 58.52 | 51.02 | 58.54 | 46.8 |
| Avengers | 62.81 | 71.56 | 52.58 | 60.89 | 53.68 | 60.30 | 51.3 |
| **Ours** | | | | | | | |
| **MoT (ours)** | 63.14 | 72.15 | 52.95 | 60.35 | 54.08 | 60.53 | 51.8 |

**Legend:** ⬛ column best; ⬛ fastest (lowest) time.

**Out-of-distribution (OOD).** Table 2 shows accuracy on PREALGEBRA, MBPP, and C-EVAL. MoT achieves the best average (47.92), exceeding AVENGERS (46.56) by $+\mathbf{2.92}\%$, ROUTERDC (45.85) by $+\mathbf{4.52}\%$, and the strongest single-model baseline (Dolphin-2.9-LLaMA-3-8B; 43.95) by $+\mathbf{9.03}\%$. MoT is best on all three OOD tasks and maintains runtime on par with AVENGERS (38.1 vs. 37.9 minutes). We attribute the stronger OOD generalization to latent-space collaboration via interaction layers: the primary expert can integrate hidden states from heterogeneous peers, going beyond single/multi-model routing or output-level aggregation.

## 4.3 ABLATIONS

**Number of interaction layers ($Q$).** We vary $Q \in \{2, 4, 6, 8\}$ while keeping all other settings fixed (Sec. 4). As shown in Fig. 6a, accuracy improves consistently as $Q$ increases. ID average rises from 56.50 at $Q{=}2$ to 60.53 at $Q{=}8$, while OOD average improves from 44.20 to 47.92 over the same range. The trend shows clear benefits from increased collaboration via more interaction layers, but gains diminish beyond $Q{=}6$, reflecting a trade-off between added computation (and parameters) from more interaction layers and incremental accuracy improvements. Refer Appendix E for details.

**Insertion depth of interaction layers.** We next study where to insert $Q{=}4$ interaction layers within each expert: *Shallow* (earliest half of layers), *Intermediate* (middle half), *Deep* (latest half), and a *Uniform* control (even spacing). Results in Fig. 6b show that deeper placement yields slightly stronger performance (58.65 ID avg., 46.23 OOD avg.), compared to shallow or intermediate placements. The uniform control (58.37 ID avg., 46.00 OOD avg.) remains competitive, suggesting that while

[1]Note: Avengers requires an offline calibration stage that clusters examples and associates each cluster with a subset of experts using labeled or unlabeled data, thus, it is not strictly training-free

Table 2: Out-of-distribution results (accuracy %).

| Method | PreAlgebra | MBPP | C-EVAL | Avg | Time (m) |
|---|---|---|---|---|---|
| **Base models** | | | | | |
| Mistral-7B | 24.80 | 37.90 | 46.43 | 36.38 | 31.3 |
| MetaMath-Mistral-7B | 39.15 | 37.74 | 45.17 | 40.69 | 30.6 |
| Zephyr-7B-Beta | 20.78 | 31.14 | 44.87 | 32.26 | 32.7 |
| Chinese-Mistral-7B | 18.48 | 29.64 | 48.44 | 32.19 | 32.9 |
| Dolphin-2.6-Mistral-7B | 29.28 | 44.86 | 45.10 | 39.75 | 28.4 |
| Meta-LLaMA-3-8B | 27.67 | 43.02 | 52.01 | 40.90 | 27.9 |
| Dolphin-2.9-LLaMA-3-8B | 39.72 | 47.34 | 44.80 | 43.95 | **27.6** |
| **Ensembles / routers** | | | | | |
| Voting | 39.03 | 41.60 | 48.50 | 43.04 | 205.4 |
| CosineClassifier | 36.97 | 38.48 | 47.77 | 41.07 | 33.0 |
| ZOOTER | 34.44 | 41.10 | 44.95 | 40.16 | 31.6 |
| LoraRetriever | 35.36 | 43.12 | 52.01 | 43.50 | 31.2 |
| RouterDC | 38.81 | 46.80 | 51.93 | 45.85 | 32.6 |
| Avengers | 38.95 | 48.11 | 52.63 | 46.56 | 37.9 |
| **Ours** | | | | | |
| **MoT (ours)** | **39.89** | **48.56** | **55.32** | **47.92** | 38.1 |

Table 3: Ablations on MoT size and primary expert cross-attention. Accuracy (%) on ID and OOD benchmarks. Inference time is wall-clock minutes.

| | In-Distribution (ID) | | | | | | Out-of-Distribution (OOD) | | | | |
|---|---|---|---|---|---|---|---|---|---|---|---|
| Method | MMLU | GSM8K | CMMLU | ARC-C | HumanEval | Avg | PreAlg. | MBPP | C-EVAL | Avg | Time (m) |
| **Ablations** | | | | | | | | | | | |
| MoT (base) | **63.14** | **72.15** | **52.95** | **60.35** | **54.08** | **60.53** | **39.89** | **48.56** | **55.32** | **47.92** | 51.8 |
| MoT (small) | 61.12 | 69.80 | 51.40 | 58.90 | 50.12 | 58.27 | 38.70 | 45.62 | 52.01 | 45.44 | **35.5** |
| MoT w/o CrossAttn | 62.71 | 69.84 | 51.23 | 58.41 | 50.12 | 58.06 | 38.14 | 47.02 | 53.71 | 46.29 | 49.5 |

later-layer collaboration is marginally more effective, uniform placement offers a simple and robust default strategy. Refer Appendix F for details.

**Lightweight MoT.** To examine the efficiency–performance trade-off, we evaluate a smaller MoT variant with reduced capacity: $Q=6$ stacks, shared dimension $d_s=1024$, and router hidden size 128. This configuration substantially reduces parameter count and training cost compared to the default ($Q=8$, $d_s=2048$, router hidden 768), while keeping the same number of attention heads ($H=8$). As shown in Table 3, the lightweight MoT maintains strong accuracy on both ID and OOD tasks, achieving only a modest drop relative to the full configuration, while reducing wall-clock inference time by ~32%. This demonstrates that MoT remains effective even under smaller parameter budgets. For parameter counts, see Appendix H.

**Removing primary expert cross-attention.** A natural question is whether MoT's gains stem merely from the additional parameters in the interaction layers, rather than from latent-space collaboration. To test this, we train a variant where the projection adapters are retained but the cross-attention module in the primary expert is removed, eliminating inter-expert latent exchange while keeping a similar parameter budget. As shown in Table 3, this variant (*MoT w/o CrossAttn*) yields noticeably lower accuracy on both ID and OOD tasks compared to the full MoT. This confirms that the performance gains are not simply due to added parameters: the cross-attention interaction in the primary expert is critical for enabling effective latent-space collaboration across heterogeneous experts.

**Effect of training loss components.** We ablate the contribution of each training loss in MoT by incrementally adding entropy ($\mathcal{L}_{ent}$), balance ($\mathcal{L}_{bal}$), and consistency ($\mathcal{L}_{con}$) regularizers to the base language modeling (LM) objective. As shown in Table 4, each component improves performance, with entropy regularization stabilizing routing, balance loss promoting even expert utilization, and consistency loss enhancing inter-expert alignment. Together, these losses yield a cumulative improvement of **+7.02%** over the LM-only baseline.

### 4.4 ANALYSIS

**Robustness to LLM loss during inference.** In real deployments, model servers may become unavailable due to hardware or network issues, making robustness to expert loss a critical property.

Table 4: Ablation of training loss components on ID benchmarks. Accuracy (%) and incremental gains.

| Configuration | Loss Components | Accuracy | Gain |
|---|---|---|---|
| **Loss ablations** | | | |
| LM Only | Language Modeling (LM) | 52.42 | – |
| LM + Ent | LM + Entropy Regularization (Entropy) | 54.36 | +1.94 |
| LM + Ent + Bal | LM + Entropy + Load Balancing (Balance) | 57.29 | +2.93 |
| LM + Ent + Bal + Con | LM + Entropy + Balance + Routing Consistency | **59.44** | +2.14 |
| **Cumulative Gain** | – | – | **+7.02** |

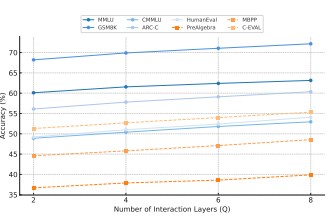
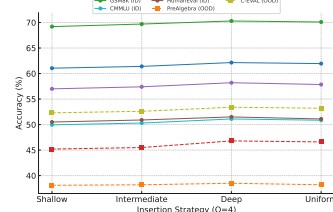
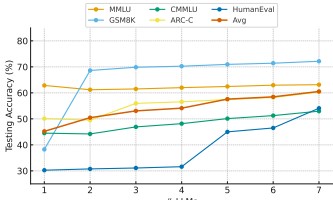

(a) **Interaction layers ($Q$).** Accuracy vs. $Q$ on ID (solid) and OOD (lighter). Gains saturate beyond $Q=6$.

(b) **Insertion depth.** Accuracy of shallow, intermediate, deep, and uniform placement of $Q=4$ layers across ID/OOD tasks.

(c) **Number of experts.** Accuracy on ID tasks as the pool grows from 1 to 7 experts with top-$K=3$ routing.

Figure 6: **MoT ablations.** (a) Effect of interaction layers, (b) insertion depth, and (c) number of experts. All variants follow the same training setup as Sec. 4.

Table 5: Robustness to expert loss during inference (accuracy %). Each row drops one expert from the 7-expert pool at test time, while using the same trained router.

| Variant | MMLU | GSM8K | CMMLU | ARC-C | HumanEval | Avg |
|---|---|---|---|---|---|---|
| **All experts vs. single drop** | | | | | | |
| All experts | **63.14** | **72.15** | **52.95** | **60.35** | **54.08** | **60.53** |
| w/o Mistral-7B | 62.40 | 71.70 | 52.50 | 59.80 | 53.40 | 59.96 |
| w/o MetaMath-Mistral-7B | 62.15 | 71.40 | 52.25 | 59.30 | 53.00 | 59.62 |
| w/o Zephyr-7B-Beta | 61.95 | 71.10 | 52.10 | 58.95 | 52.70 | 59.36 |
| w/o Chinese-Mistral-7B | 62.25 | 71.35 | 52.35 | 59.20 | 53.10 | 59.65 |
| w/o Dolphin-2.6-Mistral-7B | 62.30 | 71.50 | 52.40 | 59.40 | 53.05 | 59.73 |
| w/o Meta-LLaMA-3-8B | 61.85 | 71.00 | 51.85 | 58.85 | 52.50 | 59.21 |
| w/o Dolphin-2.9-LLaMA-3-8B | 61.70 | 70.85 | 51.80 | 58.70 | 52.35 | 59.08 |

We evaluate MOT by removing one expert at inference time while keeping the router and interaction layers unchanged from the original 7-expert training. Table 5 reports ID performance when each expert is dropped in turn. Across all settings, accuracy decreases only marginally (average $-0.57$ compared to full MOT), confirming that the framework gracefully tolerates expert loss. This robustness arises from the fact that multiple experts are routed per query and the primary expert can still integrate information from the remaining active peers through interaction layers. Thus, MOT can maintain high performance even under partial system failures.

**Effect of the number of experts.** We study scaling by enabling subsets of the expert pool (1 to 7). Fig. 6c shows accuracy improves steadily with more experts, despite training once on the full pool. Top-$K=3$ routing remains effective beyond three experts, underscoring MoT's ability to leverage heterogeneous expertise without retraining. Details are in Appendix G.

**Efficiency, scalability, and cost.** In our single-node, multi-GPU setup, MoT achieves essentially the same end-to-end latency as existing routing-based baselines, even when accounting for cross-expert communication. Auxiliary experts can be executed sequentially by storing only their projected hidden states, incurring only a small memory overhead under typical sequence lengths. The interaction layers

transfer only a few megabytes per step, making communication cost negligible relative to core model computation. We deliberately use a simple router to isolate the empirical gains arising specifically from latent-level collaboration. Moreover, MoT is compatible with stronger routing modules (e.g., RouterDC-style or cluster-based routers), which could further improve accuracy while preserving the same test-time budget.

## 5 CONCLUSION

We introduced *Mixture of Thoughts* (MOT), a method that combines global routing with latent-level collaboration to integrate multiple heterogeneous LLMs. By enabling fine-grained hidden-state exchange through lightweight interaction layers while keeping expert backbones frozen, MOT achieves single-pass inference that is both efficient and effective. Experiments on diverse in-distribution and out-of-distribution benchmarks show that MOT consistently outperforms baselines, achieving state-of-the-art performance for multi-LLM systems.

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

# A   IMPLEMENTATION DETAILS

Following previous work, we evaluate MoT using the Language Model Evaluation Harness [16]. Our default MoT configuration employs $Q = 8$ stacks, top-$K = 3$ routed experts, shared latent dimension $d_s = 2048$, router hidden size 768, and $H = 8$ attention heads in the interaction layers. The router input encoder is DeBERTaV3-base [17], used as a frozen sentence encoder. Training uses AdamW with learning rate $5 \times 10^{-5}$, batch size 64, gradient clipping, weight decay 0.01, and 50 epochs with 1000 warmup steps. Regularization weights are set to $\lambda_{\text{ent}} = 0.01$, $\lambda_{\text{bal}} = 0.01$, and $\lambda_{\text{con}} = 0.05$. For open-ended generation tasks we sample $M = 10$ outputs per query with temperature 0.2. We train on NVIDIA A100 80 GB GPUs and perform inference on NVIDIA RTX A6000 48 GB GPUs.

## A.1   MODELS

We evaluate on seven open-source decoder-only LLMs in the 7B–8B range:

- *Mistral family (5):* **Mistral-7B** [25] (general), **MetaMath-Mistral-7B** [54] (MetaMathQA-tuned), **Zephyr-7B-Beta** (DPO-aligned chat), **Chinese-Mistral-7B** (vocabulary-expanded, continued pretraining on Chinese corpora), and **dolphin-2.6-mistral-7b** (instruction-tuned, Cognitive Computations).
- *Llama-3 family (2):* **Llama-3-8B** [12] (general) and **dolphin-2.9-llama3-8b** (instruction-tuned, Cognitive Computations).

This pool mixes general, alignment-focused, domain-specialized, and multilingual experts while keeping scale comparable. Although the models ("experts") share some common architectures, each is distinct in their specialization/expertise. We also follow the experimental setup of RouterDC [6] to promote a fair evaluation environment.

For the voting baseline, we use one full forward pass per expert, on separate GPUs, and for open-ended tasks we take multiple samples per expert, which is why its latency is higher. This setup is analogous to heterogeneous self-consistency: each expert contributes an answer and a vote selects the final one, under a higher computational budget. See [6] which concludes similar latency results for the voting baseline when compared against routing- and ensemble-based baselines.

## A.2   DATASETS

**In-distribution.**   (i) **MMLU** [18] (57 subjects, broad knowledge); (ii) **GSM8K** [10] (grade-school math); (iii) **CMMLU** [32] (67 Chinese subjects); (iv) **ARC-Challenge (ARC-C)** [9] (multi-choice reasoning); and (v) **HumanEval** [5] (Python code generation). For GSM8K, we use its default train/test split; for the others, we randomly split 70%/30% following standard practice. The union of all training sets forms the data to train the global router and interaction layers; evaluation is on the held-out test sets.

**Out-of-distribution.**   (i) **PreAlgebra** [19] (algebraic problems); (ii) **MBPP** [2] (1,000 Python coding tasks); and (iii) **C-Eval** [22] (52 Chinese subjects across four difficulty levels). These datasets are unseen during training and used only for OOD evaluation.

## A.3   BASELINES

We benchmark against a broad spectrum of routing/aggregation strategies:

- *CosineClassifier*: multi-class routing via a cosine-similarity classifier.
- *Voting*: query all experts and take the majority output.
- *ZOOTER* [35]: supervised router trained with reward-based scoring signals.
- *LoraRetriever* [62]: routing from query clustering as a proxy for task identity.
- *RouterDC* [6]: supervised router using dual contrastive learning between queries and model embeddings.
- *Avengers* [58]: state-of-the-art training-free clustering with ensemble-style sampling and voting.

Together, these baselines span simple training-free heuristics and advanced supervised routers under a consistent evaluation protocol.

# B  TRAINING STRATEGY

We train only lightweight components—global router $r_\theta$, per-expert/per-stack forward/reverse projectors, and shared cross-attention weights—while keeping all expert backbones frozen.

**Primary objective.** For input sequence $\mathbf{x}$ and target sequence $\mathbf{y} = (y_0, \ldots, y_{T-1})$, the primary expert $m^*$ predicts next-token distributions $p_{m^*}(y_t \mid \mathbf{x}, \mathbf{y}_{<t})$. We minimize the autoregressive negative likelihood loss (NLL):

$$\mathcal{L}_{\text{LM}} = -\sum_{t=0}^{T-1} \log p_{m^*}(y_t \mid \mathbf{x}, \mathbf{y}_{<t}).$$

**Router exploration (entropy).** Let $\mathbf{z} = \text{PromptEncoder}(\mathbf{x})$ and $\mathbf{s} = r_\theta(\mathbf{z}) \in \mathbb{R}^M$ be router scores. To discourage collapse onto a few experts, we maximize the entropy of distribution $\pi_\tau = \text{softmax}(\mathbf{s}/\tau)$, implemented as a loss:

$$\mathcal{L}_{\text{ent}} = -\mathbb{E}_{\mathbf{x}}[\mathcal{H}(\pi_\tau)].$$

We use a straight-through (ST) relaxation: *hard* Top-$K$ indices are used in the forward pass; gradients flow through the *soft* $\pi_\tau$ in the backward pass. A standard choice is Gumbel-perturbed scores $\tilde{\mathbf{s}} = \mathbf{s} + \mathbf{g}$ (i.i.d. $\mathbf{g}$ Gumbel) with temperature $\tau$ [24, 36], yielding stable SoftTopK relaxations while preserving discrete routing at inference.

**Load balancing.** Within a batch of size $B$, let $f_m = \frac{1}{B} \sum_{i=0}^{B-1} \mathbf{1}[m \in \mathcal{I}_{\text{active}}^{(i)}]$ denote the activation frequency of expert $m$. We penalize the squared coefficient of variation to encourage uniform utilization [43, 13]:

$$\mathcal{L}_{\text{bal}} = \left(\frac{\text{std}(\mathbf{f})}{\text{mean}(\mathbf{f})}\right)^2, \qquad \mathbf{f} = (f_0, \ldots, f_{M-1}).$$

**Routing-consistency (stability under small routing changes).** Discrete Top-$K$ selections can vary due to stochastic perturbations; we regularize for stability by sampling two *independent* relaxed selections for the same input. Concretely, draw $\mathbf{g}_0, \mathbf{g}_1 \sim \text{Gumbel}^M$ and form $\tilde{\mathbf{s}}_j = (\mathbf{s} + \mathbf{g}_j)/\tau$ for $j \in \{0, 1\}$. We use *hard* Top-$K$ from $\tilde{\mathbf{s}}_0$ for the forward pass (primary distribution $p_t^{(0)}$), and in parallel we compute a distribution $p_t^{(1)}$ using experts selected via $\tilde{\mathbf{s}}_1$. We minimize token-wise KL divergence between the two distributions over the target tokens:

$$\mathcal{L}_{\text{con}} = \frac{1}{2T} \sum_{t=0}^{T-1} \left( \text{KL}(p_t^{(0)} \| p_t^{(1)}) + \text{KL}(p_t^{(1)} \| p_t^{(0)}) \right).$$

**Total objective.** The full loss combines the above terms:

$$\mathcal{L}_{\text{total}} = \mathcal{L}_{\text{LM}} + \lambda_{\text{ent}} \mathcal{L}_{\text{ent}} + \lambda_{\text{bal}} \mathcal{L}_{\text{bal}} + \lambda_{\text{con}} \mathcal{L}_{\text{con}}.$$

We use AdamW with weight decay, gradient clipping, and warmup. Weights $(\lambda_{\text{ent}}, \lambda_{\text{bal}}, \lambda_{\text{con}})$ are fixed; the defaults used in our experiments are $\lambda_{\text{ent}} = 0.01$, $\lambda_{\text{bal}} = 0.01$, $\lambda_{\text{con}} = 0.05$.

## B.1  NOTES

During training, the following hold true:

- All experts receive the same input sequence per training example.
- The global router selects top K experts and marks one as primary.
- Only the primary expert's logits are used in the language modeling loss.
- All active experts update their interaction-layer projectors and router-related parameters; backbones are frozen.
- Non-primary experts do not perform cross-attention, only their forward and reverse projections are computed.

---

**Algorithm 1** MoT Training (single batch step)

---

1: **Inputs:** batch $(\mathbf{x}, \mathbf{y})$, router $r_\theta$, projectors $\Theta_{\text{proj}}$, cross-attn $\Theta_{\text{attn}}$, frozen experts $\{\mathcal{E}_m\}_{m=0}^{M-1}$
2: Encode prompt: $\mathbf{z} \leftarrow \text{PromptEncoder}(\mathbf{x})$; $\quad \mathbf{s} \leftarrow r_\theta(\mathbf{z})$
3: Sample $\mathbf{g}_0 \sim \text{Gumbel}^M$, form $\tilde{\mathbf{s}}_0 = (\mathbf{s} + \mathbf{g}_0)/\tau$, select $\mathcal{I}_{\text{act}} = \text{TopK}(\tilde{\mathbf{s}}_0)$
4: Run active experts with interaction layers; primary $m^*$ produces distributions $\{p_t^{(0)}\}_{t=0}^{T-1}$; compute $\mathcal{L}_{\text{LM}}$
5: Compute router entropy loss $\mathcal{L}_{\text{ent}} = -\mathcal{H}(\text{softmax}(\mathbf{s}/\tau))$
6: Compute load balance loss $\mathcal{L}_{\text{bal}} = \big(\text{std}(\mathbf{f})/\text{mean}(\mathbf{f})\big)^2$ with $\mathbf{f} = (f_0, \ldots, f_{M-1})$
7: Sample $\mathbf{g}_1 \sim \text{Gumbel}^M$, form $\tilde{\mathbf{s}}_1 = (\mathbf{s} + \mathbf{g}_1)/\tau$, select $\mathcal{I}'_{\text{act}} = \text{TopK}(\tilde{\mathbf{s}}_1)$
8: Run secondary active experts $\mathcal{I}'_{\text{act}}$ with interaction layers; obtain $\{p_t^{(1)}\}_{t=0}^{T-1}$; compute $\mathcal{L}_{\text{con}}$
9: Combine: $\mathcal{L}_{\text{total}} = \mathcal{L}_{\text{LM}} + \lambda_{\text{ent}}\mathcal{L}_{\text{ent}} + \lambda_{\text{bal}}\mathcal{L}_{\text{bal}} + \lambda_{\text{con}}\mathcal{L}_{\text{con}}$
10: Update trainable parameters $\{r_\theta, \Theta_{\text{proj}}, \Theta_{\text{attn}}\}$ (experts frozen) using AdamW

---

## C  INFERENCE

MoT inference proceeds in two phases: *prefill* and *decode*. The router selects a sparse set of experts; all selected experts advance their own token streams to maintain latent trajectories, while the *primary* expert produces the final output sequence. Standard causal masking is used in the cross-attention computed within the interaction layers.

**Prefill.** Given input $\mathbf{x}$, compute $\mathbf{z} = \text{PromptEncoder}(\mathbf{x})$, scores $\mathbf{s} = r_\theta(\mathbf{z})$, select $\mathcal{I}_{\text{act}} = \text{TopK}(\mathbf{s})$, and choose the primary $m^* = \arg\max_{m \in \mathcal{I}_{\text{act}}} s_m$. Run a forward pass of all active experts on $\mathbf{x}$ with interaction layers enabled. Cache standard backbone KV pairs and the interaction-layer KV projections at every stack for each active expert. Each active expert emits its next token; the primary token is appended to the output sequence. All selected experts receive the same prompt and prefill in parallel.

**Decode.** At step $t \geq 0$, each active expert $m$ receives its own previous token (the primary expert receives the last output token). Using KV caches, the tokens traverse through the stacks. At each interaction layer, every active expert updates its KV cache; the primary expert's query token attends over the concatenated KV from all active experts to update its hidden state. New tokens are produced for all active experts; only the primary token is appended to the output. Decoding continues until the primary emits `<EOS>` or the maximum sequence length is reached. If a non-primary expert emits `<EOS>` earlier, its subsequent tokens are treated as padding and are masked in the interaction layers. Each expert maintains its own KV cache and its own token stream. The primary expert's token is the final output token. Non-primary active experts continue decoding only to supply updated hidden states to the primary experts' interaction layers.

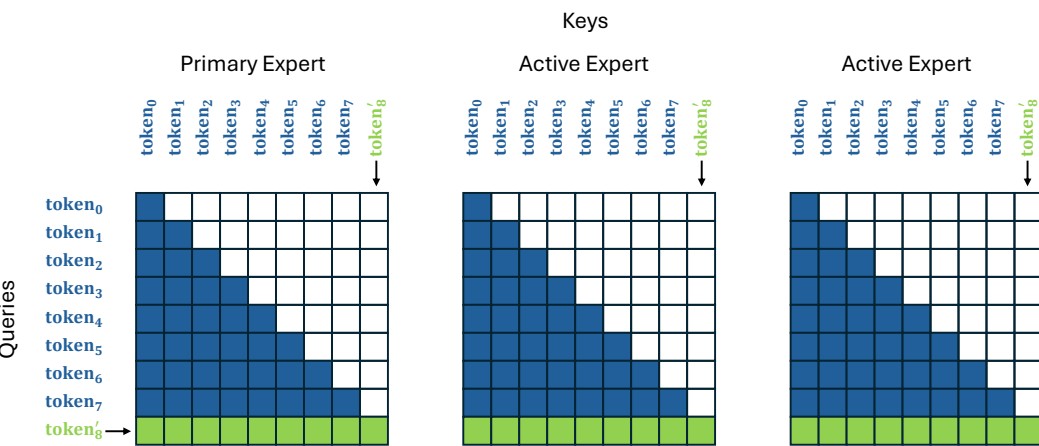

Figure 7: Attention mask used for the cross-attention computation in the interaction layers. The cross-attention is computed between queries from the primary expert and keys/values from the active experts (including the primary expert). The generated token from each active expert after prefill (shown in green) add to the KV cache for the interaction layer.

---

**Algorithm 2** MoT Inference (single sequence)

---

1: **Input:** prompt $\mathbf{x}$, experts $\{\mathcal{E}_m\}_{m=0}^{M-1}$ (frozen), router $r_\theta$, projectors $\Theta_{\text{proj}}$, cross-attn $\Theta_{\text{attn}}$, max length $T_{\max}$
2: **Route:** $\mathbf{z} \leftarrow \text{PromptEncoder}(\mathbf{x})$; $\mathbf{s} \leftarrow r_\theta(\mathbf{z})$; $\mathcal{I}_{\text{act}} \leftarrow \text{TopK}(\mathbf{s})$; $m^* \leftarrow \arg\max_{m \in \mathcal{I}_{\text{act}}} s_m$
3: **Prefill:** run all $m \in \mathcal{I}_{\text{act}}$ on $\mathbf{x}$ with interaction layers; cache backbone KV and interaction-layer KV at each stack; obtain next-token logits for each active expert; append the primary token to output $\mathbf{y}$
4: Initialize step $t \leftarrow 0$; mark all $m \in \mathcal{I}_{\text{act}}$ as `alive`
5: **while** $m^*$ is `alive` **and** $|\mathbf{y}| < T_{\max}$ **do**
6:     **for** each $m \in \mathcal{I}_{\text{act}}$ **do**
7:       **if** $m$ is `alive` **then**
8:         feed previous token of expert $m$ through its stacks using KV caches
9:         at each interaction layer: update KV cache for expert $m$; if $m = m^*$, also form the query token using the current primary token and attend over concatenated KV from all active experts (padding masked)
10:         produce next token for expert $m$ (for $m^*$ this is the output token)
11:         **if** token is `<EOS>` and $m \neq m^*$ **then**
12:           mark $m$ as `dead`
13:         **end if**
14:       **else**
15:         treat token as padding for $m$ (no compute; masked in interaction)
16:       **end if**
17:     **end for**
18:     append the primary token to $\mathbf{y}$; if `<EOS>` for $m^*$ then stop
19:     $t \leftarrow t + 1$
20: **end while**
21: **Return** $\mathbf{y}$

---

# D   RESULTS ON AIME

Table 6: **AIME results (standard training configuration).** Accuracy (%, higher is better)

| Method | AIME acc. |
|---|---|
| **Baselines** | |
| MetaMath-Mistral-7B | 24.0 |
| Dolphin-2.9-LLaMA-3-8B | 22.7 |
| LLaMA-3-8B-Instruct | 18.9 |
| Mistral-7B-Instruct-v0.2 | 17.4 |
| Zephyr-7B-beta | 15.2 |
| Chinese-Mistral-7B-Instruct | 12.8 |
| Majority vote (all experts) | 24.8 |
| RouterDC | 24.3 |
| LoraRetriever | 24.5 |
| ZOOTER | 24.7 |
| Avengers | 25.4 |
| **Ours (evaluation = OOD)** | |
| **MoT (ours)** | **26.0** |

# E    RESULTS FOR NUMBER OF INTERACTION LAYERS

Table 7: Effect of the number of stacks $Q$ (uniformly spaced interaction layers) on in-distribution (ID) and out-of-distribution (OOD) accuracy (%, higher is better). All runs keep the expert pool, router, and decoding settings fixed; only $Q$ varies.

| $Q$ | In-Distribution (ID) | | | | | | Out-of-Distribution (OOD) | | | |
| --- | --- | --- | --- | --- | --- | --- | --- | --- | --- | --- |
| | MMLU | GSM8K | CMMLU | ARC-C | HumanEval | Avg | PreAlg. | MBPP | C-EVAL | Avg |
| 2 | 60.10 | 68.20 | 48.90 | 56.10 | 49.20 | 56.50 | 36.70 | 44.60 | 51.30 | 44.20 |
| 4 | 61.55 | 69.90 | 50.40 | 57.80 | 51.00 | 58.33 | 37.90 | 45.80 | 52.70 | 45.47 |
| 6 | 62.40 | 71.05 | 51.80 | 59.10 | 52.40 | 59.35 | 38.60 | 47.10 | 54.00 | 46.57 |
| **8** | **63.14** | **72.15** | **52.95** | **60.35** | **54.08** | **60.53** | **39.89** | **48.56** | **55.32** | **47.92** |

# F RESULTS FOR INTERACTION LAYER INSERTION DEPTH

Table 8: Effect of interaction-layer insertion depth with $Q=4$ stacks. *Shallow:* earliest half of layers; *Intermediate:* middle half; *Deep:* latest half; *Uniform:* evenly spaced control. Accuracy in % (higher is better).

| Insertion | In-Distribution (ID) | | | | | | Out-of-Distribution (OOD) | | | |
| | MMLU | GSM8K | CMMLU | ARC-C | HumanEval | Avg | PreAlg. | MBPP | C-EVAL | Avg |
|---|---|---|---|---|---|---|---|---|---|---|
| Shallow | 61.05 | 69.20 | 49.95 | 57.00 | 50.50 | 57.74 | 38.10 | 45.20 | 52.30 | 45.20 |
| Intermediate | 61.40 | 69.70 | 50.30 | 57.40 | 50.90 | 58.14 | 38.20 | 45.50 | 52.60 | 45.43 |
| Deep | **62.15** | **70.30** | **51.10** | **58.20** | **51.50** | **58.65** | **38.50** | **46.80** | **53.40** | **46.23** |
| Uniform (control) | 61.95 | 70.10 | 50.85 | 57.85 | 51.10 | 58.37 | 38.20 | 46.60 | 53.20 | 46.00 |

# G    RESULTS FOR NUMBER OF EXPERTS

Table 9: Performance of sequential MoT-style model compositions. Reported metrics are accuracy (%) on standard benchmarks. The final column is the average across all tasks.

| Model Composition | #LLMs | MMLU | GSM8K | CMMLU | ARC-C | HumanEval | Avg |
|---|---|---|---|---|---|---|---|
| Mistral-7B | 1 | 62.85 | 38.25 | 44.50 | 50.10 | 30.25 | 45.20 |
| + MetaMath-Mistral-7B | 2 | 61.20 | 68.60 | 44.20 | 49.50 | 30.75 | 50.45 |
| + Zephyr-7B-Beta | 3 | 61.50 | 69.85 | 46.90 | 55.95 | 31.10 | 53.06 |
| + Chinese-Mistral-7B | 4 | 62.00 | 70.25 | 48.15 | 56.55 | 31.60 | 54.11 |
| + Dolphin-2.6-Mistral-7B | 5 | 62.45 | 70.95 | 50.10 | 57.45 | 45.00 | 57.59 |
| + Meta-Llama-3-8B | 6 | 62.95 | 71.40 | 51.25 | 58.20 | 46.50 | 58.46 |
| + Dolphin-2.9-Llama3-8B | 7 | **63.14** | **72.15** | **52.95** | **60.35** | **54.08** | **60.53** |

## H PARAMETER OVERHEAD

For $M$ experts with hidden sizes $\{d_0, \ldots, d_{M-1}\}$, $Q$ stacks (and thus $Q$ interaction layers), shared latent dimension $d_s$, and a frozen prompt encoder that outputs a $d_z$-dimensional vector. The router has two layers with hidden width $h_r$ producing $M$ scores. We count the additional (trainable) parameters due to MoT (omitting biases and layer normalization parameters for simplicity).

**Router.** A 2-layer MLP $d_z \rightarrow h_r \rightarrow M$:

$$\boxed{\#\theta_{\text{router}} \;=\; d_z h_r \;+\; h_r M}$$

**Per-expert forward/reverse projectors (per stack).** At each interaction layer $q \in \{0, \ldots, Q-1\}$ and for each expert $m$, we use a forward projector $d_m \times d_s$ and a reverse projector $d_s \times d_m$:

$$\#\theta_{\text{proj per layer}} \;=\; \sum_{m=0}^{M-1} \big(d_m d_s + d_s d_m\big) \;=\; 2 d_s \sum_{m=0}^{M-1} d_m,$$

$$\boxed{\#\theta_{\text{proj total}} \;=\; Q \cdot \left(2 d_s \sum_{m=0}^{M-1} d_m\right)}$$

**Shared cross-attention per interaction layer.** We parameterize multi-head attention with four dense projections in the shared space: $W_Q, W_K, W_V, W_O \in \mathbb{R}^{d_s \times d_s}$:

$$\#\theta_{\text{attn per layer}} \;=\; 4 d_s^2, \qquad \boxed{\#\theta_{\text{attn total}} \;=\; Q \cdot 4 d_s^2}$$

**Total overhead.** Summing the three components:

$$\boxed{\#\theta_{\text{total}} \;=\; \underbrace{d_z h_r + h_r M}_{\text{router}} \;+\; \underbrace{Q \cdot \left(2 d_s \sum_{m=0}^{M-1} d_m\right)}_{\text{proj. (all experts, all stacks)}} \;+\; \underbrace{Q \cdot 4 d_s^2}_{\text{cross-attn (all stacks)}}}$$

The added parameters scale (i) linearly with the number of experts via the projector terms, weighted by their hidden sizes $d_m$, (ii) linearly with the number of stacks $Q$, and (iii) quadratically with the shared latent size $d_s$ through the cross-attention block.

Table 10: Parameter counts of individual models used in composition. Values are reported in billions (B).

| Model | Parameters (B) |
|---|---|
| Mistral-7B-v0.1 | 6.30 |
| MetaMath-Mistral-7B | 6.30 |
| Zephyr-7B-Beta | 6.30 |
| Dolphin-2.6-Mistral-7B | 6.30 |
| Chinese-Mistral-7B-v0.1 | 6.56 |
| Meta-Llama-3-8B | 6.99 |
| Dolphin-2.9-Llama3-8B | 6.99 |
| **Total** | **45.75** |

Table 10 lists the frozen backbone models used in our system (totaling 45.75B parameters). On top of this pool, MoT introduces only lightweight trainable components. Table 11 reports the additional parameters due to the router and interaction layers across different stack depths. MoT with $Q = 8$ is the base configuration used with shared dimension $d_s = 2048$, router hidden size $h_r = 768$, Top-$K = 3$. We also experiment with smaller variants $Q = 2, 4, 6$, and an MoT (small) configuration that uses reduced dimensions $Q = 6$, $d_s = 1024$, $h_r = 128$, Top-$K = 3$. MoT adds only 0.8% to 3.4% additional parameters in comparison to the size of the frozen expert pool.

Table 11: Additional trainable parameters for MoT configurations (in **billions**, B). Overhead is computed with respect to the frozen base pool of experts (45.75B; see Table 10).

| Config | Interaction (B) | Router (B) | Total Add (B) | Overhead % |
|---|---|---|---|---|
| MoT ($Q$=2) | 0.386 | 0.001 | 0.387 | 0.85% |
| MoT ($Q$=4) | 0.772 | 0.001 | 0.773 | 1.69% |
| MoT ($Q$=6) | 1.158 | 0.001 | 1.159 | 2.53% |
| MoT ($Q$=8) | 1.544 | 0.001 | 1.545 | 3.38% |
| MoT (small) | 0.466 | $<$0.001 | 0.466 | 1.02% |

## I   MEMORY AND COMMUNICATION OVERHEAD

**Memory overhead in asynchronous sequential inference.**   MoT does not require synchronous execution across experts. Only the primary expert performs cross-attention; non-primary experts can be evaluated asynchronously and sequentially, in the same spirit as RouteLLM/Avengers-style deployments. Each auxiliary expert computes its hidden states, projects them into the shared latent space, and the resulting projected hidden states are stored for later use by the primary expert within its interaction layers.

For an auxiliary expert, the stored projected hidden states have size

$$\text{Overhead} \;=\; Q \cdot h \cdot \big(L_{\text{prefill}} + L_{\text{gen}}\big) \cdot \text{bitwidth},$$

where $Q$ is the number of interaction layers (stacks), $h$ is the hidden size, $L_{\text{prefill}}$ and $L_{\text{gen}}$ are the prefill and generation lengths, and bitwidth $= 2$ bytes (fp16/bf16). Using representative values $L_{\text{prefill}} = 256$ and $L_{\text{gen}} = 128$ (total 384 tokens), the total overhead per auxiliary expert is shown in Table 12.

Table 12: Projected hidden-state memory overhead per auxiliary expert as a function of the number of interaction stacks $Q$ and hidden size $h$, assuming $L_{\text{prefill}} = 256$, $L_{\text{gen}} = 128$, and fp16/bf16.

| $Q$ | $h$=1024 | $h$=2048 |
|---|---|---|
| 2 | 1.57 MB | 3.14 MB |
| 4 | 3.14 MB | 6.29 MB |
| 6 | 4.71 MB | 9.43 MB |
| 8 | 6.29 MB | 12.57 MB |

Even in the heaviest configuration ($Q = 8$, $h = 2048$), a single auxiliary expert contributes only $\approx 12.6$ MB of stored projections, i.e., less than 0.09% of a 7B model's fp16 parameter memory ($\approx 14$ GB). With $K = 3$ routed experts, the total projection overhead from the two auxiliary experts is about 25 MB.

At the stack level, a 7B model with 32 layers corresponds to roughly 1.75 GB per stack (for $Q = 8$). If auxiliary experts are executed sequentially, the primary expert only requires an additional $\approx 3.12$ MB of projected data from the two auxiliary experts at each stack. This overhead scales linearly with sequence length; even for sequence lengths on the order of $10^4$, the memory overhead remains under 100 MB. For very long-context settings, this overhead becomes more significant, which we view as a limitation of the current design and an interesting direction for future work.

**Communication overhead in synchronous multi-GPU MoT inference.**   For synchronous deployment, we place each expert on a separate GPU. This is the setup used for the latency measurements in Table 18 and Table 7. At each interaction layer during prefill, the primary expert must receive projected hidden states from the $K - 1$ auxiliary experts. The amount of data transferred is

$$(K - 1) \cdot L_{\text{prefill}} \cdot h \cdot \text{bitwidth}.$$

Using representative values $L_{\text{prefill}} = 256$, $h = 2048$, and bitwidth $= 2$ bytes, this corresponds to 1.05 MB per auxiliary expert per interaction layer. For $K = 3$, the total transfer per interaction layer is therefore 2.1 MB.

During generation, the per-token communication size is even smaller and independent of sequence length:

$$(K - 1) \cdot h \cdot \text{bitwidth}.$$

Table 13: Representative transfer time per interaction layer for 2.1 MB of projected activations (two auxiliary experts, $K = 3$).

| Interconnect | Transfer Volume | Transfer Time |
|---|---|---|
| PCIe 4.0 (x16, $\sim$ 32 GB/s) | 2.1 MB | $\approx 0.06$ ms |
| NVLink 3 ($\gtrsim$ 100 GB/s) | 2.1 MB | $\approx 0.02$ ms |

For $h = 2048$ and fp16, this is about $4$ KB per auxiliary expert, or roughly $8$ KB total when $K = 3$.

Even when accounting for interconnect startup latencies (on the order of microseconds in typical PCIe and NVLink deployments), the communication time at these message sizes remains negligible relative to the overall prefill and decode latency. This is consistent with our empirical results: MoT exhibits essentially the same end-to-end inference time as routing-based baselines such as Avengers under our shared hardware configuration.

Standard pipelining can further hide communication latency. For example:

1. each expert computes its current stack $s$;

2. the auxiliary experts asynchronously send projected hidden states to the primary expert, which performs the interaction-layer cross-attention;

3. in parallel, auxiliary experts proceed to stack $s+1$ and prepare the next set of projections.

Designing and evaluating such system-level scheduling strategies is an appealing avenue for future work.

**Inference setup and GPU memory.** Our inference setup follows standard multi-LLM practice, as in RouterDC. Each expert is hosted on its own RTX A6000 (48 GB) GPU, and the router dynamically selects the top-$K$ experts per query. We use fp16 (or bf16) without activation checkpointing, CPU offloading, or quantization; the router and interaction layers are lightweight relative to the expert backbones. This configuration ensures that the additional memory and communication overhead introduced by MoT remains small compared to the underlying model and hardware costs.

## J  TRAINING DETAILS

This section summarizes the training settings used in our runs. All expert backbones are frozen; only the router, per-expert forward/reverse projectors, and shared cross-attention weights are trained. Details are covered in Tables 14 through 17.

**Expert pool (frozen).**

- `mistralai/Mistral-7B-v0.1`

- `meta-math/MetaMath-Mistral-7B`

- `HuggingFaceH4/zephyr-7b-beta`

- `cognitivecomputations/dolphin-2.6-mistral-7b`

- `itpossible/Chinese-Mistral-7B-v0.1`

- `meta-llama/Meta-Llama-3-8B`

- `cognitivecomputations/dolphin-2.9-llama3-8b`

**Implementation details.** In our implementation, only the primary expert executes cross attention in the interaction layers. For each selected auxiliary expert, we first compute its hidden states, apply the forward projection into the shared latent space, and store the resulting projected activations. The primary expert then reads these stored projections at its interaction layers and performs cross attention over them. This design does not require all experts to run in lockstep: auxiliary experts can be evaluated one after another on a single device, storing only their projected hidden states for later use by the primary expert, or they can be placed on separate devices that asynchronously send their projections to the primary expert, which gathers them before applying cross attention. We fix top–$K$ = 3 for our main experiments. In all main experiments we set K = 3, which we found to provide a

Table 14: MoT base configuration used in training. The base configuration matches our main experiments.

| Setting | Value |
|---|---|
| Stacks ($Q$) | 8 |
| Top-$K$ | 3 |
| Shared latent dim ($d_s$) | 2048 |
| Interaction heads | 8 |
| Router hidden size ($h_r$) | 768 |
| Router temperature ($\tau$) | 1.0 |
| Dropout (interaction layers) | 0.1 |
| Prompt encoder (frozen) | `microsoft/deberta-v3-base` |

Table 15: Optimization hyperparameters.

| Hyperparameter | Value | Hyperparameter | Value |
|---|---|---|---|
| Epochs | 50 | Batch size | 64 |
| Grad. accumulation | 1 | Learning rate | $5 \times 10^{-5}$ |
| Weight decay | 0.01 | Warmup steps | 1000 |
| Grad clip (L2) | 1.0 | Max length (tokens) | 512 |
| Precision | FP16 | Seed | 42 |

good balance between accuracy and cost. In 6, we show that as the expert pool grows from 1 to 7 experts, routing to 3 experts remains effective and accuracy continues to improve. A systematic study of K is an interesting direction for future work.

**Hardware details.** Each expert is hosted on its own RTX A6000 GPU in our main experiments. We use fp16, no quantization, no CPU offloading, no activation checkpointing for inference. The router is lightweight and resides with the primary.

## K    ADDITIONAL EXPERIMENTS

**Motivation.** A potential concern with multi-benchmark evaluations (Section 4) is overlap between training and evaluation data. Such leakage could artificially inflate performance on both ID and OOD tasks. To address this, we conduct additional experiments following the same setup as before, but with separation between training and evaluation datasets. This ensures that our results fairly demonstrate the advantages of MoT over prior approaches.

**Training datasets.** For training we use the following datasets: commonsense and multi-choice QA (35% of steps; HellaSwag, CommonsenseQA, ARC-Challenge), math and quantitative reasoning (25%; GSM8K, SVAMP), code generation (25%; MBPP and a 10k subset of APPS), and reading comprehension (15%; SQuAD v2).

**Sampling and batching.** Percentages above denote the share of *optimizer updates* allocated to each domain: at each update we sample a domain with probabilities $(0.35, 0.25, 0.25, 0.15)$ and then draw the batch from a dataset within that domain using temperature-based sampling (temperature $= 2.0$). We use a sequence length of $2048$ and a batch size of $64$.

**Evaluation.** We train and evaluate using the same seven 7B to 8B experts. Training setting is identical to that in Sec. K. We report accuracy for benchmarks that are *not* used for training.

**Discussion.** Under the standard training configuration, **MoT (Base)** remains the strongest overall. Compared to AVENGERS, MoT improves the baseline-defined ID average by **+0.42** (55.97 vs. 55.55) and the OOD average by **+1.95** (47.35 vs. 45.40). Importantly, these gains are achieved even though *all evaluation datasets are OOD for MoT*, since none were used during training. Per-benchmark improvements are consistent: MMLU (+0.55), CMMLU (+0.10), HumanEval (+0.60), PreAlgebra

Table 16: Loss weights used during training.

| Component | Weight |
|---|---|
| Entropy regularization ($\lambda_{\text{ent}}$) | 0.01 |
| Load balancing ($\lambda_{\text{bal}}$) | 0.01 |
| Routing consistency ($\lambda_{\text{con}}$) | 0.05 |

Table 17: Decoding/evaluation settings.

| Parameter | Value | Parameter | Value |
|---|---|---|---|
| Beam size / samples | 10 | Temperature | 0.2 |
| Top-$p$ | 0.9 | Max new tokens | 256 |
| Return sequences | 1 | | |

(+1.10), and C-EVAL (+2.80). This highlights that MoT generalizes robustly and outperforms baselines, despite being evaluated in a stricter zero-shot setting where even their ID datasets are OOD for MoT.

## L    USE OF LARGE LANGUAGE MODELS

We used a large language model to assist with grammar, phrasing, LaTeX formatting, and routine table editing. All suggested changes were reviewed and edited by the authors. No proprietary or sensitive data were provided to the model.

## M    LIMITATIONS

Our experiments reflect realistic budget and hardware constraints: we use a modest pool of frozen experts with limited parameter counts and interaction depth to keep compute, memory, and latency practical. The MoT design itself is compatible with larger scales; future work will systematically scale the expert pool and sizes, experiment with interaction layer placement and training technique, and co-design systems (e.g., fused projector/attention kernels) to map quality–cost Pareto fronts across broader domains.

Because of our fixed computational budget, our experiments have limited exploration over top K and expert pool size, especially for larger and more diverse models. Current experiments are with 7–8B experts, thus, scaling to larger models is an interesting point of future work. Additionally, long-context scenarios may incur higher memory overhead due to stored projections.

## N    THEORETICAL

Our focus is primarily architectural and empirical: we demonstrate that latent-space interaction between frozen heterogeneous experts improves performance under a single-pass decoding budget. A rigorous theoretical analysis of cross-model latent interaction is left to future work. From a theoretical perspective, MoT can be viewed as a sparse mixture-of-experts (MoE) model instantiated at the level of full pretrained decoders. Classical MoE formulations consider a set of experts $\{f_e\}_{e=1}^{E}$ and a gating network $g(x)$ that produces input dependent mixture weights, yielding a prediction of the form

$$f(x) \;=\; \sum_{e=1}^{E} g_e(x)\, f_e(x),$$

with analysis typically focusing on conditional computation, specialization of experts on different regions of the input space, and improved approximation of heterogeneous target functions. In MoT, the global router plays the role of a sparse gating network that selects a top–K subset of experts and designates a primary expert for each input, so that different experts specialize on different domains and query types. However, rather than combining experts only at the level of final logits, MoT allows the primary expert to attend over the latent representations of its peers through cross attention in a

Table 18: **Additional experiments (standard training configuration).** Accuracy (%, higher is better) on evaluation benchmarks not used for MoT training. For baselines, "ID" refers to datasets included in their training; "OOD" refers to held-out datasets. For MoT, *all* evaluation datasets are OOD since none were used for training.

| Method | Baselines: In-Distribution (ID) | | | | Baselines: Out-of-Distribution (OOD) | | |
|---|---|---|---|---|---|---|---|
| | MMLU | CMMLU | HumanEval | Avg | PreAlg. | C-EVAL | Avg |
| **Baselines** | | | | | | | |
| Best single model (Dolphin-2.9-LLaMA-3-8B) | 59.46 | 44.72 | 49.39 | 51.19 | 39.72 | 44.80 | 42.26 |
| RouterDC | 60.80 | 51.50 | 50.20 | 54.17 | 38.30 | 51.70 | 45.00 |
| Avengers | 62.05 | 52.00 | 52.60 | 55.55 | 38.70 | 52.10 | 45.40 |
| **Ours (all evaluation = OOD)** | | | | | | | |
| **MoT (Base)** | **62.60** | **52.10** | **53.20** | **55.97** | **39.80** | **54.90** | **47.35** |

small number of interaction layers. This corresponds to a content dependent mixture of intermediate features, rather than a convex mixture of outputs. The interaction layers also connect MoT to representation learning and feature fusion. The projected hidden states of auxiliary experts can be viewed as specialized feature views of the input, trained on different data distributions or tasks. Cross attention in the primary expert then implements a learned retrieval mechanism over these views: for each token and layer, the primary expert queries the shared latent space and selects which auxiliary features to integrate. This is closely related to ideas in multi view and modular representation learning, where combining complementary feature subspaces can improve generalization, and to theoretical analyses of MoE which show that routing can partition the input space into regions where different expert representations are most informative. In this sense, MoT can be interpreted as a latent space MoE that reuses frozen expert representations while learning a lightweight router and interaction mechanism that decide when and how to compose them. The term "thoughts" in Mixture of Thoughts refers to latent space interaction, rather than what it means in applications like chain-of-thought.

