# OpenReview forum: "Mixture of Thoughts: Learning to Aggregate What Experts Think, Not Just What They Say"
_ICLR.cc/2026/Conference — Submitted to ICLR 2026_

### Official Review · Reviewer_Q4vY · 2025-10-17

**Soundness:** 2
**Presentation:** 3
**Contribution:** 2
**Rating:** 2
**Confidence:** 4

**Summary:**

Authors propose a method to aggregate pretrained LLM decoders, Mixture of Thoughts (MoT). The idea is to let a primary expert “look at” the hidden states of other active experts and use that information thought cross attention. For each query, a router selects the top-K experts and designates a primary expert. The authors insert interaction layers that project each model’s hidden states into a shared latent space; the primary then performs cross-attention over the selected peers’ latents. All experts are frozen; only the router and interaction layers are trained with a joint objective.

While I like the idea of “merging” pertained models, there are several drawbacks in the proposed implementation as well as in the evaluation setup (see Weaknesses). I think that the marginal improvements on downstream tasks do not justify the overhead and complexity of the proposed approach.

**Strengths:**

- The paper introduces a new way to enable communication between pretrained LLM decoders by exchanging latent representations rather than final outputs. This latent-level interaction is conceptually interesting and bridges ideas from model ensembles and mixture-of-experts architectures.
- It is well written and easy to follow.

**Weaknesses:**

The main drawback is that injecting interaction layers makes inference synchronous across experts. One of the main benefits of using a pool of models is the ability to aggregate their results asynchronously, but this design removes that advantage.

Let us consider two cases:
1. On device inference with limited memory (we can fit only one model)
In prior works such as RouteLLM and Averagers, we can process the prompt, select experts, and then sequentially: load an expert, perform the computation, save the output, and repeat this process k times. In the proposed approach, however, this becomes problematic because intermediate hidden states must be stored to allow the primary expert to attend to them later. This introduces additional memory reads and writes, which can further stress an already memory-bound inference setup.

2. All experts placed on different devices:
In this scenario, interaction layers require a gather operation by the primary expert. While the exchanged hidden states are small and the bandwidth cost is minimal (even over Ethernet I suppose), this design introduces a synchronization point. In heterogeneous setups, where devices or models have varying latencies, these synchronization points could accumulate and lead to noticeable overhead.

I would appreciate seeing a table similar to Table 10 in the Appendix, but focusing on overhead rather than parameter counts. In particular, it would be helpful to include:

- communication overhead when models are placed on different devices with different type of interconnect
- memory overhead
- latency for generation ( authors provide average time across downstream tasks but I would be more interested in latency for prompts of different length)

A table summarizing these measurements would make it much easier to understand the trade-offs.
If the authors can explicitly demonstrate that generation speed is not hindered by the additional communication and provide a detailed analysis of memory and bandwidth requirements (see Questions), the proposed method would be significantly more convincing.

**Questions:**

Inference is reported to be conducted on an RTX A6000 GPU with 48 GB of memory. The authors state that $k = 3$, which implies that three $7$B models (approximately $21$B active parameters) are used simultaneously. Assuming bfloat16 precision, the model weights alone would require roughly 42GB of memory, leaving very limited space for the rest (KV caches, router, etc). This suggests that inference must have been distributed across multiple devices or involved CPU offloading.
Could the authors please clarify the inference setup, specifically:
- whether experts parallelism was used,
- whether any activation checkpointing, quantization, or CPU offloading was employed

to fit the models into 48 GB?

---

> ### Author Response · Authors · 2025-11-25
>
> ## **Memory Overhead in Asynchronous Sequential Inference**
>
> We appreciate the reviewer’s insightful comments. MoT does not require synchronous execution across experts. Only the primary expert performs cross-attention. All non-primary experts can be evaluated asynchronously and sequentially, exactly as in RouteLLM/Avengers: each auxiliary expert computes its hidden states, projects them into the shared latent space, and these projected hidden states are stored for later computation by the primary expert within its interaction layer.
>
> For an auxiliary expert, the stored projected hidden states have size:
>
> $$
> \text{Overhead} = Q \cdot h \cdot (L_{\text{prefill}} + L_{\text{gen}}) \cdot \text{bitwidth},
> $$
>
> where \(Q\) is the number of interaction layers, $h$ is hidden size, $L_{\text{prefill}}$ and $L_{\text{gen}}$ are prefill and generation lengths, and bitwidth = 2 bytes (fp16/bf16). Using representative values $(L_{\text{prefill}} = 256$, $L_{\text{gen}} = 128$, total = 384 tokens), the total overhead per auxiliary expert is:
>
> | Q | h=1024 | h=2048 |
> |---|--------|---------|
> | 2 | 1.57 MB | 3.14 MB |
> | 4 | 3.14 MB | 6.29 MB |
> | 6 | 4.71 MB | 9.43 MB |
> | 8 | 6.29 MB | 12.57 MB |
>
> Even in the heaviest configuration $(Q = 8, h = 2048\)$, one auxiliary expert contributes only **12.6 MB** of stored projections, i.e., **<0.09%** of a 7B model’s fp16 parameter memory (≈14 GB). With $K = 3$, the total overhead is ≈25 MB.
>
> At the stack level, a 7B model with 32 layers corresponds to ≈1.75 GB per stack (Q=8). If executed sequentially, the primary expert would only require an additional **3.12 MB** of data from the two auxiliary experts (K=3). This overhead scales linearly with sequence length; even for sequence lengths on the order of $10^4$, the memory overhead remains under 100 MB.
>
> We agree that for long-context inference, this overhead becomes substantial—this is a current limitation of MoT that we will address in upcoming work.
>
> ---
>
> ## **Communication Overhead in Synchronous Multi-GPU MoT Inference**
>
> For synchronous deployment, we place each expert on a separate GPU. We analyze the scenario where experts are located on distinct devices—the latency results reported in Table 1 follow this setting.
>
> At each interaction layer during prefill, the primary expert must receive projected hidden states from the \(K-1\) auxiliary experts. This requires transferring:
>
> $$
> (K-1) \cdot L_{\text{prefill}} \cdot h \cdot \text{bitwidth}.
> $$
>
> Using representative values $(L_{\text{prefill}} = 256$, $h = 2048$, bitwidth = 2 bytes), this gives a per-auxiliary-expert, per-stack overhead of **1.05 MB**. For $K = 3$, the total per-interaction-layer overhead is **2.1 MB**.
>
> **Transfer times for 2.1 MB:**
>
> | Interconnect | Transfer Volume | Transfer Time |
> |--------------|------------------|----------------|
> | PCIe 4.0     | 2.1 MB           | 0.06 ms        |
> | NVLink 3     | 2.1 MB           | 0.02 ms        |
>
> During generation, per-token communication is even smaller and independent of sequence length:
>
> $$
> (K-1)\cdot h \cdot \text{bitwidth}.
> $$
>
> For $h = 2048$ and fp16, this is **4 KB per expert**, or **8 KB total**.
>
> Even when accounting for PCIe and NVLink startup latencies—typically ≈1–3 µs for PCIe transactions and under 1 µs for NVLink link-level traversal [NVIDIA GPUDirect RDMA; NVIDIA DGX A100 Architecture]—the communication cost at these message sizes remains negligible and does not change the conclusions of the above analysis.
>
> Compared to both the prefill and decode latency, these overheads are negligible. This is reflected in our results in Table 1, where MoT has **essentially identical inference time to Avengers**.
>
> Standard pipelining can further hide communication latency:
>
> 1. each expert computes its current stack $s$;
> 2. auxiliary experts asynchronously send projected hidden states to the primary expert, which performs the interaction-layer collaboration;
> 3. simultaneously, auxiliary experts proceed to stack $s+1$ and prepare the next projections.
>
> Such system-level scheduling optimizations are promising directions for future work.
> We will also include a detailed breakdown of prefill vs. generation latency in the revised draft.
>
> ---
>
> ## **Inference Setup and GPU Memory**
>
> We appreciate the reviewer’s question and clarify that our inference setup follows the standard multi-LLM configuration used in prior work (such as RouterDC). Each expert is hosted on its own **RTX A6000** GPU, and the router dynamically selects the top-K experts per query.
>
> We do **not** use activation checkpointing, CPU offloading, quantization, or other memory-saving techniques. Each expert runs as a standard fp16/bf16 model on a single GPU.
>
> We will add these details to the draft for clarity.

---

### Official Review · Reviewer_bsPj · 2025-10-25

**Soundness:** 3
**Presentation:** 4
**Contribution:** 3
**Rating:** 6
**Confidence:** 3

**Summary:**

The authors propose Mixture of Thoughts (MOT), a method for latent-level collaboration among heterogeneous experts using a global router. For each query, a lightweight router selects the top-K experts and a primary expert. Interaction layers with adapters and cross-attention are added to each expert, allowing their hidden states to be integrated into the primary expert, while keeping the expert backbones frozen. A joint training objective optimizes both the router and interaction layers.

**Strengths:**

1. The paper is clearly written and easy to follow.
2. The use of both ID and OOD datasets effectively demonstrates the method's generalizability.
3. An expert loss experiment provides evidence for the method's robustness.

**Weaknesses:**

1. The experts chosen (from Llama and Mistral families) seem similar. It would be insightful to test experts from more diverse families to assess generalization.
2. In "Removing Primary Expert Cross-Attention," comparing against the optimal performance of all experts with LoRA, rather than just removing cross-attention, could better highlight its impact.
3. The model without auxiliary loss (52.42%) performs lower than ensembles/routers (53-60%). This suggests auxiliary loss is important. Could other ensembles/routers improve with it?

**Questions:**

1. I'm curious about how the choice of the Top-k parameter influences the accuracy of the proposed method. Could varying Top-k lead to noticeable changes in results, and what trade-offs should practitioners be aware of?
2. The standard deviation is not reported in this paper.

---

> ### Author Response · Authors · 2025-11-25
>
> ## **Clarity on the expert pool being “similar”**
>
> Our experts are all 7B–8B models from the LLaMA and Mistral architectural families. However, they are intentionally quite heterogeneous in terms of training data, specialization, and behavior. As summarized in Table 9, the pool includes:
>
> - general-purpose chat models (Mistral-7B, Zephyr-7B-Beta, Meta-Llama-3-8B, Dolphin-2.9-Llama3-8B)
> - a math-specialized model (MetaMath-Mistral-7B)
> - a coding-specialized model (Dolphin-2.6-Mistral-7B)
> - a Chinese-specialized model (Chinese-Mistral-7B-v0.1)
>
> So while they share similar parameter counts and base architectures, they are diverse in domain expertise and training corpora, which is exactly the setting MoT targets. We will clarify this in the main text to avoid the impression that the pool consists of several nearly identical copies of the same model. We also agree that extending to more architecturally diverse families (for example, including Qwen or Qwen2 models) is a natural direction for future work.
>
> Our experimental setup follows prior work such as RouterDC, which allows for a direct and fair comparison.
>
> ---
>
> ## **On the “Removing primary expert cross-attention” ablation and the suggestion to compare to “all experts with LoRA”**
>
> The aim of the “MoT w/o CrossAttn” ablation is to isolate the effect of latent interaction itself under a comparable parameter budget. In this variant, we keep the projection adapters but remove the primary expert cross-attention, so there is no inter-expert latent exchange—only extra parameters. Table 3 shows that this variant has noticeably lower accuracy than full MoT on both ID and OOD tasks, which directly answers the question: *are the gains just coming from more parameters, or from actual latent-space collaboration?*
>
> The comparison suggested by the reviewer—LoRA fine-tuning each expert on the same data—targets a different setting from the one we study. In our evaluation, we follow standard practice in [6] and related routing work: the “single-model” baselines are off-the-shelf specialist models that have already been fully fine-tuned or domain-adapted by their original authors (e.g., MetaMath-Mistral, Dolphin-2.9, Zephyr). Introducing an additional LoRA stage would modify the experts themselves and shift the comparison toward parameter adaptation rather than collaboration among frozen heterogeneous models. We will clarify this distinction in the revised version.
>
> For this reason, we believe the current ablation is the appropriate control for the specific hypothesis we are testing (impact of cross-attention under similar parameter budget). We will explicitly state this goal in the paper for clarity.
>
> ---
>
> ## **On the auxiliary losses and the LM-only MoT baseline**
>
> The LM-only MoT variant (52.42% in Table 4) is weaker than several ensemble/router baselines that lie around 53–60%. This is expected: LM-only MoT uses only the language-modeling loss, without the routing-stabilization or load-balancing regularizers.
>
> The main takeaway in Table 4 is the incremental effect of each component:
>
> - Adding the **entropy term** improves performance (54.36) by stabilizing routing.
> - Adding the **balance term** further improves performance (57.29) by preventing collapse onto a few experts.
> - Adding the **consistency loss** yields the full MoT configuration in this ablation setting (59.44), which is competitive with or better than the ensemble/router methods.
>
> We agree that the auxiliary losses are important for MoT’s performance. However, they are not generic add-ons that can be directly applied to all ensemble or routing methods. Our loss terms are tailored to a global top-K router with shared latent interaction, whereas training-free baselines (Voting, CosineClassifier, Avengers) do not train a router, and supervised routers such as RouterDC already employ their own training schemes. We will revise the text to clarify that Table 4 is an ablation of the MoT training objective.
>
> ---
>
> ## **On the choice of Top-K and its influence on accuracy**
>
> All main experiments in the current draft fix **Top-K = 3**. The sensitivity study in Fig. 6c varies the **number of experts in the pool** (from 1 to 7) while keeping K fixed, and shows that K = 3 routing remains effective as more experts become available, with accuracy improving steadily.
>
> We do not yet include a dedicated sweep over different Top-K values. We agree this is an important practical question. Due to limited computational resources and time constraints, we were unable to train MoT under multiple K settings. In the revision, we will:
>
> - add an explicit Top-K ablation on a representative subset of benchmarks, and
> - provide a clear discussion of the expected trade-offs.
>
> We will also clarify the role of K in the current experiments to avoid any ambiguity.
>
> ---

---

> > ### Author Response · Authors · 2025-11-25
> >
> > ## **On standard deviations**
> >
> > We do not report standard deviations or confidence intervals in the current version. Following prior work in routing and multi-LLM collaboration, our experiments use fixed seeds and a single run per configuration, as repeated evaluation is computationally expensive. However, we agree that reporting variability would strengthen the experimental section. In the revision, we will include a small set of repeated runs to estimate variance on a representative subset of benchmarks.
> >
> > ---
> >
> > Thank you for the constructive feedback and for the positive assessment of the paper’s clarity and robustness experiments. Your comments help us refine the presentation of the expert pool, clarify the intent behind our ablations, and strengthen the motivation for additional analysis of Top-K and variance reporting.

---

> > > ### Comment · Reviewer_bsPj · 2025-11-25
> > >
> > > I maintain my rating, and I look forward to the authors fulfilling their commitment to further refinements and providing additional experimental analysis in the subsequent revision.

---

### Official Review · Reviewer_MLY1 · 2025-10-29

**Soundness:** 2
**Presentation:** 2
**Contribution:** 2
**Rating:** 4
**Confidence:** 2

**Summary:**

This paper proposes Mixture of Thoughts (MoT), a novel and efficient framework for collaboratively leveraging multiple specialized open-source large language models (LLMs). Unlike prior approaches that rely on routing, iterative aggregation, or model merging, MoT introduces lightweight routers and cross-attention-based interaction layers in the latent space, enabling selected expert models to exchange information during a single forward pass. With frozen pre-trained experts and only the router and interaction layers trained, MoT achieves state-of-the-art performance across in-distribution and out-of-distribution benchmarks.

**Strengths:**

1. Innovative Methodology. The paper proposes the Mixture of Thoughts (MoT) framework, introducing a latent-level collaboration mechanism in multi-LLM systems. It enables information fusion across experts through a lightweight interaction layer, serving as a significant extension to existing routing or ensemble approaches.
2. Efficiency and Performance Balance. It achieves state-of-the-art performance with single-pass inference, runtime comparable to routing-based baselines, and avoids the overhead associated with iterative aggregation.
3. Comprehensive Experimental Design. The method is evaluated on five in-domain tasks such as MMLU and GSM8K, and three out-of-domain tasks including MBPP and C-Eval, with comparisons against strong baselines like RouterDC, Avengers, and ZOOTER.
4. Thorough Empirical Analysis. The paper presents ablation studies on the number of interaction layers, insertion positions, number of experts, and expert dropout, demonstrating a degree of robustness.

**Weaknesses:**

1. Limited Gains in In-domain Settings. MoT is supervised trained on in-domain data using a joint loss to optimize both the router and interaction layers, while the baseline Avengers requires no training. However, the absolute improvements over Avengers are only +0.23% on in-domain tasks and +1.36% on out-of-domain tasks, which appear marginal given the additional training cost.
2. Lack of Scalability Analysis. The paper reports that generation time is comparable to Avengers, but MoT requires simultaneous loading of k LLMs, and decoding involves inter-model communication and memory overhead. Avengers, on the other hand, supports fully parallel inference without communication. The paper does not verify whether MoT maintains linear scalability under common multi-GPU inference setups, where communication cost could become a bottleneck.
3. Weak Theoretical Justification. The paper lacks theoretical analysis explaining why interaction in the latent space leads to performance gains.

**Questions:**

1. Scalability Issue. Although the authors claim that generation time is comparable to Avengers, MoT involves communication of hidden states between experts, whereas Avengers allows for full parallelization without inter-model communication. When using large models that require multi-GPU setups, communication costs may significantly hinder MoT’s efficiency. The authors should provide a detailed analysis of communication, memory, and other scalability-related overhead to validate MoT’s efficiency in multi-GPU settings.
2. Is the Performance Gain Worth the Training Cost? MoT is supervised trained on in-domain data, while Avengers achieves competitive performance without any training. Given the training overhead and implementation complexity of MoT, do the modest performance improvements justify the cost?

---

> ### Author Response · Authors · 2025-11-25
>
> Thank you very much for the detailed review and for highlighting both the methodological novelty and the breadth of the experiments. We address each concern below and will incorporate these clarifications into the revised version.
>
> ## **Limited gains in in-domain settings vs. Avengers and training cost**
>
> We agree that the raw averages (+0.23 points on ID and +1.36 points on OOD tasks) look modest at first glance. We would like to clarify why we view these gains as meaningful given the strength of the baselines and the small training overhead.
>
> - **Avengers is a strong baseline**, yet MoT matches or improves it on 4 of 5 in-domain tasks and is strictly better on all three OOD datasets, so the gains are broad rather than driven by a single outlier task. Also, Avengers is not truly training-free: the method requires creating clusters from data (we used our training data).
>
> - **MoT adds only 0.8–3.4 percent trainable parameters** on top of the 45.75B frozen experts (Table 10), since we train only the global router and interaction layers. This is a small training overhead relative to pretraining and represents a one-time cost that can be leveraged across many downstream queries.
>
> - **Inference time is essentially identical to Avengers** under our shared setup (ID: 60.53 vs 60.30 accuracy in 51.8 vs 51.3 minutes; OOD: 47.92 vs 46.56 in 38.1 vs 37.9 minutes), yielding consistent accuracy gains without extra inference cost.
>
> - **We deliberately use a simple MLP router** to emphasize and isolate the performance gains from the MoT architecture itself.
>
> - **Our experts are relatively small (7–8B)** because of computational limits, which likely underestimates MoT’s full potential. We expect performance to improve with larger or more diverse experts, but training and evaluating such setups is currently infeasible for us, similar to prior work such as RouterDC that also operates on smaller ensembles.
>
> Overall, MoT achieves consistent 1–3 point improvements over a strong baseline on in-domain tasks and larger gains on OOD benchmarks, while incurring only a small additional training cost and no extra inference overhead. Together with the fact that MoT is, to our knowledge, the first method to realize latent-level collaboration between heterogeneous experts, this makes it a relevant and promising paradigm for multi-expert model design.
>
> ---
>
> ## **Multi-GPU setups**
>
> Our current experiments are conducted in a setting where the active experts for a query fit on one node, and we report end-to-end wall-clock time that already includes any interaction overhead. Under this regime, MoT and Avengers show very similar latency, which suggests that the cross-expert interaction cost is small relative to the base model computation. In particular:
>
> - MoT routes each query to top-K experts and only the primary expert performs cross-attention over the projected hidden states of the top-K experts at a small number of interaction layers. Non-primary experts only apply lightweight projections. This limits both the number of experts that participate and the number of layers where communication occurs.
>
> - The communicated tensors are standard activations (sequence length by hidden size), and interaction layers are relatively shallow. In a multi-GPU environment where each expert is already using tensor or pipeline parallelism, the additional communication required to gather projected states at the primary expert is of the same order as the costs already present in standard MoE or tensor-parallel deployments.
>
> - Since the top-K experts vary by query, one can profile routing patterns for the target workload, identify experts that frequently co-occur among the top-K, and co-locate those experts on the same node or high-bandwidth group to reduce cross-device communication. We leave such system-level co-location strategies as an interesting direction for future work.
>
> We provide a detailed breakdown of the memory and communication costs in response to reviewer Q4vY. Our analysis and latency results demonstrate that MoT remains scalable across a wide range of multi-GPU settings, wherein the memory and/or communication overhead related to the shared latent hidden states is minimal. Our revised draft will include additional profiling results across multi-GPU settings to support this claim.

---

> > ### Author Response · Authors · 2025-11-25
> >
> > ## **Theoretical justification for latent-space interaction**
> >
> > We thank the reviewer for this suggestion. Our focus in this work is empirical and architectural rather than theoretical. We view a rigorous theory for cross-model latent interaction as an interesting and complex open problem. Our ongoing work attempts to address this; for the present version, we can strengthen the intuition and connection to prior work.
> >
> > MoT can be viewed as a sparse mixture of experts where (i) the router learns to select a small set of heterogeneous experts and designate a primary expert, and (ii) the primary expert aggregates a learned weighted combination of their hidden states via cross-attention. This is closely related to known benefits of MoE and representation fusion: different experts specialize on different regions of the input space, and cross-attention allows the primary model to query these specialized representations in a content-dependent way rather than committing to a single fixed expert.
> >
> > In the revision, we will make clear that our contributions are empirical, and we will add a brief discussion section connecting MoT to existing theoretical insights on MoE and representation learning while explaining the mechanism intuitively.
> >
> > ---
> >
> > ## **Is the performance gain worth the training cost compared to Avengers?**
> >
> > MoT and Avengers occupy slightly different points in the design space. Avengers performs offline calibration to assign each model to a cluster, routing each query to the most suitable cluster based on similarity. MoT instead requires a small amount of supervised data and light training to jointly optimize routing and interaction. The resulting tradeoff is:
> >
> > - **Training overhead:** MoT trains only lightweight components (router and interaction layers) that represent less than 4 percent additional parameters relative to the frozen expert pool in our experiments.
> >
> > - **Inference overhead:** At test time, MoT has essentially the same throughput and latency as Avengers, as both query a small number of experts in parallel and decode in a single pass. Our results show that MoT consistently improves over Avengers in both ID and OOD settings under the same inference budget.
> >
> > - **Flexibility:** Because the backbone experts are frozen and only the small router and interaction layers are trained, adapting MoT to a new domain or expert pool requires retraining only these lightweight components, which is much cheaper than fine-tuning or re-merging experts.
> >
> > Avengers is also not entirely training-free: it requires clustering experts using calibration data (in our case, derived from the training set) to map each input to the appropriate clusters and associated experts. In this sense, both methods incur an offline adaptation cost, but MoT delivers larger and more consistent gains across eight benchmarks while keeping inference cost on par with these “training-free” baselines. We will clarify this cost profile and the role of calibration in Avengers more explicitly in the paper.
> >
> > We are grateful for the reviewer’s careful reading and for raising these points about gains, scalability, and theory. They help us refine both the framing and the technical discussion of MoT, and we will incorporate the clarifications and additional analyses described above in the revision.

---

### Official Review · Reviewer_eUqQ · 2025-11-02

**Soundness:** 2
**Presentation:** 2
**Contribution:** 3
**Rating:** 4
**Confidence:** 4

**Summary:**

This paper introduces Mixture of Thoughts (MoT), a framework to drive multiple pre-trained LLMs to perform effective collaboration. The core idea is to aggregate their internal "thought" representations through by adding some adapter blocks interleaved with the original transformer blocks. Besides, the paper also introduce a lightweight and learnable router to select a subset of experts for an input query. During inference, the router select a subset of experts with a primary expert, which integrates the representations of other experts and produces the final response. This work conducts experiments on five benchmarks, include both in-distribution and out-of-distribution setups. The results demonstrate that the proposed framework outperforms router-based baselines. Besides, ablation studies validat the key design choices of the MoT architecture.

**Strengths:**

- The central concept of aggregating latent "thoughts" rather than just outcomes is an intuitive research area of multi-agent systems. It addresses the limitation of cross-model communication, where models/agents typically communicate with text rather than some latent representations.
- The proposed framework consistently surpasses strong router-like baselines on a diverse set of benchmarks, for both in-domain and out-of-domain setups.
- Ablation studies systematically validate the contributions of key architectural components such as the number of interaction-layer number of depth. Ablation studies support the architectural designs of the proposed framework.

**Weaknesses:**

- Limited Scope of Baselines: The paper's abstract and related work section astutely categorize prior multi-LLM approaches into three paradigms: (i) routing, (ii) response-level collaboration and (iii) parameter fusion. However, the selected baselines for comparison are predominantly router-based or simple output-ensembling methods (e.g., Voting, RouterDC).  The experimental evaluation would be significantly more comprehensive if it included representative baselines from the other two categories.
- Limited Validation on Thinking-Centric Tasks: The title, "Mixture of Thoughts," sets a high expectation for deep collaboration on complex reasoning. While the chosen benchmarks like GSM8K are valuable, the evaluation would be more convincing if it included tasks that require longer chains of thought (e.g., AIME).
- Lack of Qualitative Analysis on Collaboration Dynamics: Since the paper seems to focus on aggregate the "thoughts", the paper would be strengthened by an analysis of the tought aggregation behaviors usch as cross-expert attention patterns. Such an analysis could provide direct evidence of meaningful collaboration, for instance, by showing that a primary expert dynamically attends to a math-specialized expert's hidden states when solving a math problem.
- Clarity on Training/Inference Implementation: The precise mechanics of the token-by-token generation for non-primary experts could be more explicit in the main text to improve clarity. Could the authors clarify what inputs each expert receives during training? Are all experts given the same input sentences? My current understanding is that the inputs are identical during training but differ during inference—if so, does this create a mismatch between training and inference?

**Questions:**

- The reported latency for the "Voting" baseline is much higher than one would expect from a single, parallelized forward pass, where latency should be bounded by the slowest expert. Does this implementation involve generating multiple candidate samples from each expert and then voting on the final answers?

- To better isolate the benefits of MoT's heterogeneous collaboration from the general benefits of ensembling, have you considered a baseline where the single strongest expert uses self-consistency by voting on its own multiple sampled results, using a comparable computational budget to MoT?

---

> ### Author Response · Authors · 2025-11-25
>
> Thank you for the careful reading and constructive suggestions. We appreciate that you found the idea of latent thought aggregation intuitive and the empirical gains convincing.
>
> ## **Limited scope of baselines (routing vs collaboration vs fusion)**
> Our primary design goal is to enable single-pass collaboration across a heterogeneous pool of expert LLMs. Parameter-fusion and distillation-based methods [51, 55, 14, 21, 59, 52, 47] fundamentally require homogeneous architectures and therefore cannot be compared in our heterogeneous setting. Such approaches directly merge or distill model parameters. Response-level collaboration methods (e.g., MoA [48]) also fall outside our evaluation scope: they depend on multi-turn, agentic interactions between experts, resulting in substantially higher inference cost. Crucially, Avengers [58] – the current state of the art in multi-expert collaboration – already shows that MoA-style systems significantly underperform routing-based methods, despite their higher cost.
>
> Accordingly, we compare against RouterDC, ZOOTER, and Avengers because they are the strongest and most relevant state-of-the-art baselines for single-pass multi-expert collaboration. We will clarify the rationale for selecting the baselines in the paper.
>
> ## **Limited validation on “thinking centric” reasoning tasks (e.g., AIME)**
> Our current evaluation already includes GSM8K and PreAlgebra, which require multi step chain of thought style reasoning and are standard in recent routing and multi LLM work, in part to enable a fair comparison to RouterDC and Avengers under a shared setup.
> Originally, we did not perform experiments on AIME because the smaller ~7B model sizes that we included in our experiments (due to computational budgets) do not perform well on AIME (<5-10%). That said, AIME and similar competition level math benchmarks would further stress deep reasoning, so we will include these experiments in the final paper.
>
> ## **Qualitative analysis of collaboration dynamics**
> We will add a new analysis section (with figures) that:
>
> - **Visualizes cross-attention weights** in the interaction layers for different query types.
>   For example, we will show that for math questions the primary generalist model places higher attention mass on the **MetaMath** expert in later stacks, while for Chinese questions it relies more heavily on **Chinese-Mistral**.
>
> - **Breaks down expert contributions** by measuring how often each model is selected as primary vs. auxiliary across different task groups.
>
> - **Reports an “attention entropy over experts” metric** to quantify how many experts meaningfully participate in the primary expert’s cross-attention for each domain.
>
> These additions will provide the direct evidence requested by the reviewer, demonstrating that MoT performs **meaningful, query-sensitive latent collaboration**, rather than acting as a static ensemble.

---

> ### Author Response · Authors · 2025-11-25
>
> ## **Clarity on Training and Inference Inputs for Experts**
> We appreciate the comment and will update the main text content for clarity. The precise training and inference inputs for each expert are provided in Appendix B and Appendix C; we will move a summary of these details into the main method section. For completeness, we clarify the reviewer’s questions here:
>
> - **Training inputs:**
>   For each training instance, *all* experts receive the same input sequence (prompt plus gold tokens via standard teacher forcing). The global router selects a top-K set and designates a primary expert. Only the primary expert’s logits are used in the language-modeling loss; however, all active experts update their interaction-layer projectors and router-related parameters. This is analogous to fine-tuning a single autoregressive model, except that (i) the experts’ backbone weights remain frozen, and (ii) their hidden states are coupled through the primary expert’s interaction layers.
>
> - **Inference inputs:**
>   During inference, all active experts receive the same prompt in the prefill phase. In the decode phase, each expert autoregressively consumes its *own* previously generated token stream. The primary expert’s token is taken as the final output, while non-primary experts continue decoding only to update their hidden states for use in the interaction layers.
>
> There is no discrepancy between training and inference—both follow the standard autoregressive LLM training/inference procedure. We will expand the method section to explicitly walk through these phases.
>
> In addition, we will clarify the following implementation details:
>
> - **Only the primary expert runs cross-attention in the interaction layers.** Non-primary experts apply only their forward and reverse projectors without attention. This is mentioned in Sec. 3.3, but we will make the computational implications more explicit.
>
> - **Each expert maintains its own KV cache and token stream, but only the primary expert’s logits are used to select the next output token.**
>
> This results in a per-step cost that is effectively:
> **“top-K routed experts in parallel + one cross-attention block in the primary expert,”**
> which explains why our measured latency closely matches RouterDC and Avengers.
>
> ## **Voting baseline latency**
> Regarding the latency of the Voting baseline, our implementation is correct and follows the standard evaluation protocol. For each query, we run a separate forward pass for every expert on a single GPU, and for open-ended generation tasks we additionally sample multiple candidates per expert before voting. Voting places a single GPU per model. Voting baseline is known to be significantly more computationally expensive than single-pass baselines, as also reported in RouterDC [6].
>
> ## **Self-consistency baseline**
> We agree that self-consistency is useful, but it targets a different setting. Our work studies how heterogeneous experts can collaborate under a single-pass decoding budget. Classical self-consistency repeatedly samples from the same model and votes over multiple outputs, which is computationally costly and does not use model heterogeneity. Importantly, the voting baseline we include already serves as a self-consistency analogue for heterogeneous experts: each expert contributes one pass, and the final answer is chosen by voting. This makes it more appropriate than single-model self-consistency for our setting. We will clarify that our baselines focus on multi-expert, single-pass methods rather than sampling-based extensions of a single model.
>
> ## **Wording of claims**
> Thank you for pointing out the potential ambiguity in the title. In this work, the term “thoughts” refers to the internal latent representations of each expert, not to explicit chain-of-thought text. This is standard terminology in representation-level collaboration work, where intermediate hidden states are often informally described as the model’s “thoughts” or “internal reasoning”. MoT operates entirely in this latent space, aggregating hidden states across experts inside the transformer, and does not rely on or modify chain-of-thought prompting. We will clarify this explicitly in the introduction to avoid suggesting that the method is designed specifically for very long textual chains of thought.

---

### Author Response · Authors · 2025-12-03
**Rebuttal Draft Update**

We thank all reviewers for their thoughtful feedback and constructive suggestions. Your comments significantly improved the clarity, scope, and rigor of the paper. We have **updated the draft** to address all substantive concerns (**marked in blue**), added new analyses and experiments where requested, and clarified methodological and implementation details throughout. We appreciate the time and care invested in the reviews, and we hope that the revised version satisfactorily resolves the points raised.

---

### Meta-Review · Area_Chair_sxCK · 2026-01-11

**Summary:**

I recommend rejection for this submission. Although the idea of multi-expert collaboration through latent-space interaction is conceptually interesting, it seems that the empirical gains are marginal (0.23% average improvement on in-domain tasks) to justify the added architectural complexity and training overhead, particularly when compared to training-free baselines like Avengers. I am also concerned that the lack of theoretical justification for why latent-space interaction improves performance leaves the core contribution insufficiently grounded. Although the authors provided thoughtful responses during the rebuttal and committed to revisions, I feel the fundamental cost-benefit tradeoff remains questionable, and the theoretical gap would need to be addressed before this work is ready for publication at this venue.

**Reviewer Concerns:**

The authors are commended for introducing a conceptually interesting approach to multi-expert collaboration and for providing thorough responses during the rebuttal period. However, the empirical gains remain marginal (0.23% average improvement on in-domain tasks) relative to the added architectural complexity and training requirements, and the lack of theoretical justification for why latent-space interaction improves performance (acknowledged by the authors as future work) leaves the core contribution insufficiently grounded for acceptance at this venue.

Reviewers raised substantive concerns about the modest performance improvements not justifying the training overhead compared to training-free baselines like Avengers, the absence of theoretical analysis explaining the mechanism behind latent collaboration, and questions about scalability under practical multi-GPU deployment scenarios. The authors provided detailed technical responses including memory and communication overhead calculations, and committed to adding reasoning-intensive benchmarks, cross-attention visualizations, and expanded ablations in revision. While many concerns were addressed through analysis and revision commitments, the theoretical justification remains unresolved and the fundamental cost-benefit tradeoff continues to be questioned.

**Reviewer Scores:**

The score distribution reflects divided opinions: two reviewers scored marginally below acceptance (4), one marginally above (6), and one strongly advocated for rejection (2). The most critical reviewer maintained that the marginal improvements do not justify the approach's complexity, while the most positive reviewer found the contribution acceptable pending committed revisions. Overall, the reviewer consensus does not support acceptance, with three of four reviewers at or below the acceptance threshold.

---

### Decision · Program_Chairs · 2026-01-26

Reject